# PML mutants from arsenic-resistant patients reveal SUMO1-TOPORS and SUMO2/3-RNF4 degradation pathways

Ellis G. Jaffray[1], Michael H. Tatham[1], Barbara Mojsa[1], Anna Plechanovová[1], Alejandro Rojas-Fernandez, Julio C.Y. Liu[2], Niels Mailand[2], Adel F.M. Ibrahim[1], Graeme Ball[3], Iain M. Porter, and Ronald T. Hay[1]

**Arsenic effectively treats acute promyelocytic leukemia by inducing SUMO and ubiquitin-dependent degradation of the promyelocytic leukemia (PML)–retinoic acid receptor alpha oncogenic fusion protein. However, some patients relapse with arsenic-resistant disease because of missense mutations in PML. To determine the mechanistic basis for arsenic resistance, PML$^{-/-}$ cells were reconstituted with YFP fusions of wild-type PML-V and two common patient mutants: A216T and L217F. Both mutants were resistant to degradation by arsenic but for different biochemical reasons. Arsenic did not trigger SUMOylation of A216T PML, which failed to recruit the SUMO-targeting ubiquitin ligases RNF4 and TOPORS. L217F PML did respond with increased SUMO2/3 conjugation that facilitated RNF4 engagement but failed to reach the threshold of SUMO1 conjugation required to recruit TOPORS. Thus, neither mutant accumulated the appropriate polyubiquitin signal required for p97 binding. These PML mutants have revealed a convergence of SUMO1, SUMO2/3, TOPORS, and RNF4 that facilitates the arsenic-induced degradation of PML.**

## Introduction

Acute promyelocytic leukemia (APL) is caused by a reciprocal chromosomal translocation t(15;17) that fuses the genes encoding the promyelocytic leukemia (PML) protein and the retinoic acid receptor alpha (RARA) (de Thé et al., 1991; Kakizuka et al., 1991). This generates the PML-RARA oncoprotein that deregulates transcriptional programs required for the differentiation of hematopoietic progenitor cells. Therefore, differentiation is blocked and promyelocytes accumulate causing leukemia (Grignani et al., 1993; Kwok et al., 2006; Martens et al., 2010; Mikesch et al., 2010; Tan et al., 2021).

PML is a member of the tripartite motif (TRIM) family of proteins and is also known as TRIM19. In healthy cells, the PML gene is alternatively spliced to generate multiple mRNAs that encode seven protein isoforms (I-VII) that differ in their C-terminal regions. Located in the N-terminal region of PML and present in all PML isoforms and PML-RARA fusions, the highly conserved TRIM motif is composed of a RING domain, two zinc coordinating B-boxes, and a coiled-coil that mediates dimerization (Bernardi and Pandolfi, 2007). In cells with wild-type (WT) PML genes, the protein is associated with non-membranous nuclear structures known as PML nuclear bodies, which also accumulate a variety of other proteins including small ubiquitin-like modifiers (SUMOs), p53, Daxx, and SP100. In cells containing a PML-RARA fusion protein, PML and PML-RARA are co-associated in tiny nuclear speckles (Dyck et al., 1994; Weis et al., 1994). APL was a disease with a very poor prognosis until a combination therapy consisting of arsenic trioxide (referred to herein as arsenic or As) and *all-trans* retinoic acid (Lo-Coco et al., 2013; Mi et al., 2015; Wang and Chen, 2008) was developed. These induce degradation of the PML-RARA oncogene and allow the WT version of RARA to initiate a transcriptional program that drives the accumulated promyelocytes down pathways of apoptosis and terminal differentiation, which ultimately cures the disease. Arsenic alone induces degradation of both the unfused PML and PML-RARA, identifying PML as the target of arsenic. Treatment with arsenic leads to rapid multisite modification of PML and PML-RARA with SUMO (Müller et al., 1998). SUMO-modified PML and PML-RARA then serve to recruit SUMO-targeted ubiquitin E3 ligases (STUbLs). One such STUbL, RING Finger Protein 4 (RNF4), is recruited to SUMO-modified

[1]Division of Molecular, Cell and Developmental Biology, School of Life Sciences, University of Dundee, Dundee, UK; [2]Protein Signaling Program, Novo Nordisk Foundation Center for Protein Research, University of Copenhagen, Copenhagen, Denmark; [3]Dundee Imaging Facility, School of Life Sciences, University of Dundee, Dundee, UK.

Correspondence to Ronald T. Hay: r.t.hay@dundee.ac.uk

A. Rojas-Fernandez's current affiliation is Instituto de Medicina & Centro Interdisciplinario de Estudios del Sistema Nervioso (CISNe), Universidad Austral de Chile, Valdivia, Chile. I.M. Porter's current affiliation is IMPACT Imaging Facility, Centre for Discovery Brain Sciences, University of Edinburgh, Edinburgh, UK.



PML and PML-RARA present in PML nuclear bodies via multiple SUMO interaction motifs (SIMs) in the N-terminal region of the protein (Lallemand-Breitenbach et al., 2008; Tatham et al., 2008). High local concentrations of RNF4 trigger homodimerization of the C-terminal RING domain, which is required for ubiquitin E3 ligase activity (Rojas-Fernandez et al., 2014). Two further SIM-containing STUbLs have been implicated in PML turnover: RNF111 (Arkadia) (Erker et al., 2013) and TOPORS (Liu et al., 2024), although the degree of redundancy and cooperation among the three STUbLs is unclear. Prior to degradation, ubiquitinated PML or PML-RARA is extracted by the p97/VCP segregase (Jaffray et al., 2023), which is thought to unfold and guide the modified protein to the proteasome where it is proteolytically degraded. Although most APL patients treated with arsenic are cured, a small proportion relapse and present with arsenic-resistant disease. In some cases, resistance may be a consequence of metabolic reprogramming linked to the expression of additional known oncogenes (Iaccarino et al., 2019; Madan et al., 2016), but in many patients, arsenic resistance is a result of missense mutations in PML clustered between residues L211 and S220 in B-box 2. These mutations are found not only in the PML-RARA oncogene (Chendamarai et al., 2015; Goto et al., 2011; Iaccarino et al., 2016; Liu et al., 2016; Lou et al., 2015) but also in the unfused version of PML (Iaccarino et al., 2016; Lehmann-Che et al., 2014; Zhu et al., 2014). These mutations are undetectable at initial diagnosis, suggesting that cells expressing these mutations are selected for by arsenic treatment (Alfonso et al., 2019; Balasundaram et al., 2022).

To establish the mechanistic basis for arsenic resistance in these mutants, we investigated the properties of two mutations in PML that responded differently to arsenic. While both mutations were resistant to arsenic-induced degradation, the A216T mutant was severely compromised for SUMO modification, failed to recruit TOPORS and RNF4, and as a result was not ubiquitinated. In contrast, like WT-PML, the L217F mutant was modified by SUMO2/3, recruited RNF4, and was ubiquitinated in response to arsenic. However, SUMO1 modification and TOPORS recruitment were compromised, and the ubiquitin signal was not sufficient to recruit the p97 segregase that is required to extract PML from nuclear bodies. These findings suggest the arsenic-induced PML degradation pathway may rely on both SUMO paralogs whereby SUMO1 is associated with TOPORS activity and SUMO2/3 with RNF4, and where both are required for efficient degradation of PML.

## Results

### PML mutants A216T and L217F are not degraded in response to arsenic

Mutations in PML and PML-RARA that arise in APL patients with arsenic-resistant disease are a unique research resource as these mutations are selected in vivo to be resistant to arsenic while still retaining the biological functions of PML that contribute to the disease phenotype. Most of these mutations are located to a 10–amino acid region (L211-S220) in B-box 2 of PML that encompasses, but never includes, the zinc coordinating residues C212 and C215 (Fig. S1 A). As PML is the target of arsenic

and mutations from patients with arsenic-resistant disease can also be detected in the non-rearranged allele of PML (Iaccarino et al., 2016; Lehmann-Che et al., 2014), we chose to reconstitute U2OS PML$^{-/-}$ cells with WT or mutant versions of PML-V N-terminally fused to YFP (Jaffray et al., 2023). In testing several PML variants, we found that they fell into two main groups exemplified by the most frequently identified mutations in patients: A216T and L217F. U2OS cells expressing YFP-fused forms of WT, A216T, and L217F PML-V were isolated by FACS and the derived cell lines analyzed by high-content microscopy. Images of these cells showed that while total YFP fluorescence was similar for WT, A216T, and L217F, the size and number of PML bodies per cell varied (Fig. 1 A). Quantitative analysis of thousands of cells showed that the average expression levels, measured by YFP-PML-V intensity per cell, were similar for all three PML forms (Fig. 1 B), although cells expressing A216T and L217F PML-V had fewer and larger PML bodies than WT (Fig. 1 B). These results show that A216T and L217F mutations in PML alter the morphology of PML bodies in a background lacking WT-PML. Arsenic exposure is responsible for dramatic changes to the number, morphology, and content of PML bodies (Geoffroy et al., 2010; Jeanne et al., 2010). Concentrations of arsenic >1 µM have been shown to have proteotoxic and cytotoxic effects (Kang et al., 2003). We therefore monitored the effect of 1 µM arsenic on YFP-PML body morphology for WT and the two mutants using live-cell imaging. At least nine movies were collected for each condition, and an automated, quantitative pipeline (Jaffray et al., 2023) was used to follow PML body size, number, and total YFP-PML-V fluorescence intensity over time (see Videos 1, 2, 3, 4, 5, and 6 for representative time course data). YFP-PML fluorescence images from the cells at 0 and 15 h showed a reduction in the number and total intensity of nuclear bodies in WT-PML-V–expressing cells, but not in cells expressing either A216T or L217F PML-V (Fig. 1 C). Averaged values of these metrics across all real-time analyses indicated that PML body size did not significantly change over time after arsenic treatment for any of the PML forms studied (Fig. 1 D). However, the number of bodies per cell and normalized YFP-PML-V intensity were reduced during arsenic exposure for WT-PML but not for either mutant (Fig. 1 D). After 15-h arsenic, there were a greater than fourfold reduction in the number of PML bodies per cell and a greater than threefold reduction in YFP-PML-V intensity for the WT-PML-V, while little or no change was observed for A216T and L217F mutants (Fig. 1 E), indicating that WT-PML is efficiently degraded in response to arsenic, while the A216T and L217F PML mutants are resistant to arsenic-induced degradation. Immunoblot analysis of extracts gathered from cells exposed to arsenic over 24 h showed WT-PML rapidly shifts to a higher molecular weight species before being degraded (Fig. 1 F). In contrast, the mobility of A216T PML does not appear to alter at all, while L217F PML does form lower mobility conjugates, but is not degraded after 24-h exposure to arsenic (Fig. 1 F).

### Differential modification of PML mutants with SUMO and ubiquitin

To better understand the nature of the more slowly migrating forms of PML, a protease treatment strategy of purified PML

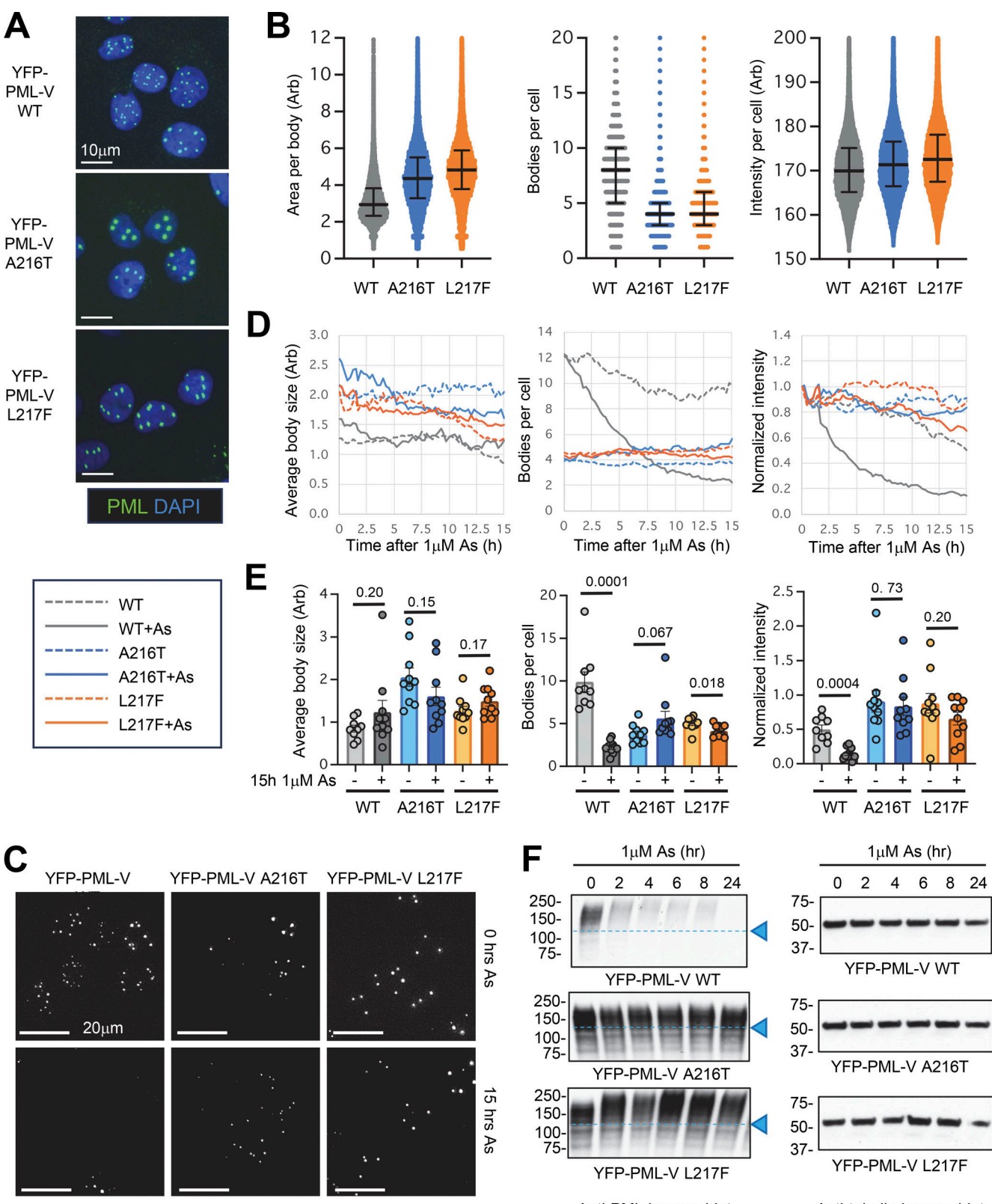

Figure 1. **A216T and L217F mutants of PML are not degraded in response to arsenic treatment. (A)** Fluorescence microscope images of the indicated PML⁻/⁻ U2OS + YFP-PML-V cell lines showing YFP (PML) and DAPI (DNA) fluorescence. **(B)** High-content imaging data summarizing total PML intensity per cell, PML body number per cell, and area per PML body for each cell line. Scatter plots are shown with median (solid line) and quartiles (error bars). Number of cells, $n$ = 38411 (WT), 35210 (A216T), and 41288 (L217F). **(C)** YFP fluorescence image stills at 0- and 15-h treatment with 1 µM arsenic from a single live-cell analysis of U2OS PML⁻/⁻ + YFP-PML-V WT, A216T, and L217F (summarized in D and E). **(D)** Average PML body size, number of bodies per cell, and PML intensity per cell over a 15-h time course of exposure to 1 µM arsenic. Intensity measurements are normalized by the t = 0 values. Fields of view per condition, $n$ = 9 or 10 with a

median of 12 cells per field. **(E)** Summary statistics for data shown in D for 15-h 1 µM arsenic treatment and untreated cells. Columns are averages with SEM error bars (fields of view, n = 9 or 10). P values are determined using unpaired, two-tailed Student's t tests using Welch's correction for unequal variances where appropriate. See Videos 1, 2, 3, 4, 5, and 6 for real-time data of a single representative field. **(F)** Anti-PML and anti-tubulin immunoblots of whole-cell extracts taken from the YFP-PML-V cell lines at the indicated times during 1 µM arsenic exposure. Source data are available for this figure: SourceData F1.

bodies was developed to broadly assess the modification status of all three YFP-PML forms (Fig. 2 A). Cells were either untreated or treated with 1 µM arsenic for 2 h and nuclear bodies purified from WT-, A216T-, and L217F YFP-PML-V–expressing cells using a YFP-specific nanobody attached to magnetic beads (Jaffray et al., 2023). Purifications were then treated with SUMO or ubiquitin-specific proteases to deconjugate these modifiers from resin-bound material (Fig. 2 B). The material released by protease treatment, and that remaining on the beads can be analyzed by immunoblotting, which provides information not only on the covalent modification status of PML, but also on that of the SUMO1, SUMO2/3, and ubiquitin species attached to PML (Fig. 2 B). Prior to PML body purification, nuclear extracts for WT, A216T, and L217F YFP-PML-V displayed the expected changes in electrophoretic mobility of PML and were almost completely depleted from nuclear extracts after YFP-PML-V purification (Fig. 2 C). Importantly, the modifications to PML were preserved during purification (Fig. 2 D, no protease). For all PML variants, treatment of the beads with SENP1 returned most of the heavier PML forms back to more rapidly migrating species consistent with being unmodified (Fig. 2 D, SENP1), indicating most of the low mobility PML is directly modified by SUMO. Treatment with USP2 had little impact on the distribution of PML species, indicating only a relatively small proportion of the modified PML is ubiquitin-conjugated (compare Fig. 2 D, no protease, with USP2). SUMO1 conjugation to any PML type, including WT, is almost undetectable in these experiments unless cells were treated with arsenic, which causes a dramatic increase in modification of YFP-PML-V WT, which is blunted for L217F PML and undetectable for A216T (Fig. 2 D, SUMO1 blot). As expected, these species disappear upon SENP1 treatment but are largely unaltered by treatment with USP2 (compare Fig. 2 D, SUMO1 blot). Unlike SUMO1, SUMO2/3 conjugates to PML were detectable prior to arsenic exposure, but only for WT and L217F variants (Fig. 2 D, SUMO2 blot). The relative increase in SUMO2/3 conjugation upon arsenic treatment is smaller than for SUMO1, and A216T PML is modified to a much lesser degree than either WT or L217F (Fig. 2 D, SUMO2/3 blot). As with SUMO1, SUMO2/3 is also completely removed by SENP1, and USP2 treatment does not result in significant changes to SUMO2/3 antibody–reactive species (Fig. 2 D, SUMO2/3 blot). Ubiquitin is present at low levels on WT, A216T, and L217F YFP-PML-V in untreated cells but upon arsenic exposure is greatly increased on WT-PML, modestly increased on L217F PML, and changes little for A216T PML (Fig. 2 D, ubiquitin blot). For both WT-PML and the L217F mutant, SENP1 treatment not only reduces the amount of ubiquitin remaining on the beads, but also causes the remaining ubiquitin-reactive species to run at a reduced molecular weight (Fig. 2 D, ubiquitin blot). This remainder is presumably ubiquitin directly conjugated to PML and will likely consist of monomeric ubiquitin, as well as ubiquitin in

polymeric chains which cannot be discriminated between here. These species were confirmed to be ubiquitin by treatment with USP2 (Fig. 2 D, ubiquitin blot).

The material released from the beads by treatment with the proteases was also analyzed by immunoblotting (Fig. 2 E). As expected, none of the treatments released YFP-PML-V (Fig. 2 E, PML blot). SENP1 treatment released low levels of SUMO1 from untreated WT and L217F YFP-PML-V samples, with none yielded from A216T purifications (Fig. 2 E, SUMO1 blot). Considerably, more SUMO1 was released from the samples derived from arsenic-treated cells, with the most being associated with WT-PML, less from L217F PML forms, and trace amounts from A216T PML (Fig. 2 E, SUMO1 blot). Notably, from WT-PML the SUMO1 released by SENP1 was in two forms, with a secondary form ~10 kDa heavier than the unconjugated SUMO1 (S1* in Fig. 2 E). These are expected to be SUMO1-conjugated by ubiquitin, as they are sensitive to USP2 (Fig. 2 E, SUMO1 blot). This species was not apparent in SENP1 elutions from L217F-PML purifications, implying ubiquitination of SUMO1 is more extensive for the WT protein. SENP1 digestion released substantial amounts of monomeric SUMO2/3 from preparations derived from untreated cells for all PML variants including A216T (Fig. 2 E, SUMO2/3 blot). Arsenic treatment induced a modest increase in SUMO2/3 conjugates on all three PML forms, and a ubiquitinated form of SUMO2/3 was detected in the SENP1-released material for both WT and L217F PML (S2* in Fig. 2 E). No ubiquitin-SUMO2/3 adducts were detected in the A216T PML purifications. These SUMO-ubiquitin species released by SENP1 were also detected using a ubiquitin antibody (Fig. 2 E, ubiquitin blot).

To assess the recruitment of SUMO1 and SUMO2/3 to PML bodies by a complementary method, we carried out immunofluorescence analysis. U2OS PML−/− + YFP-PML-V cells were either untreated or treated with arsenic for 2 h and stained with antibodies to SUMO1 or SUMO2/3. PML was detected using YFP fluorescence. In response to arsenic, SUMO1 is strongly recruited into WT-PML bodies, modestly recruited to L217F-PML bodies, but not to A216T-PML (Fig. 3 A, left). SUMO2 does accumulate in WT and L217F PML bodies after arsenic treatment, but this is not observed with A216T (Fig. 3 A, right). Quantitation of colocalization of YFP-PML WT and mutants with SUMO1 and SUMO2 (Fig. 3 B) is broadly consistent with the protease treatment analysis shown in Fig. 2.

### In response to arsenic, WT and L217F PML recruit RNF4, but the A216T mutant does not

Arsenic-induced SUMO modification of PML is responsible for the recruitment of the STUbL RNF4 that contributes to PML ubiquitination prior to degradation by the proteasome (Lallemand-Breitenbach et al., 2008; Tatham et al., 2008). To study RNF4 recruitment in response to arsenic, nuclear bodies

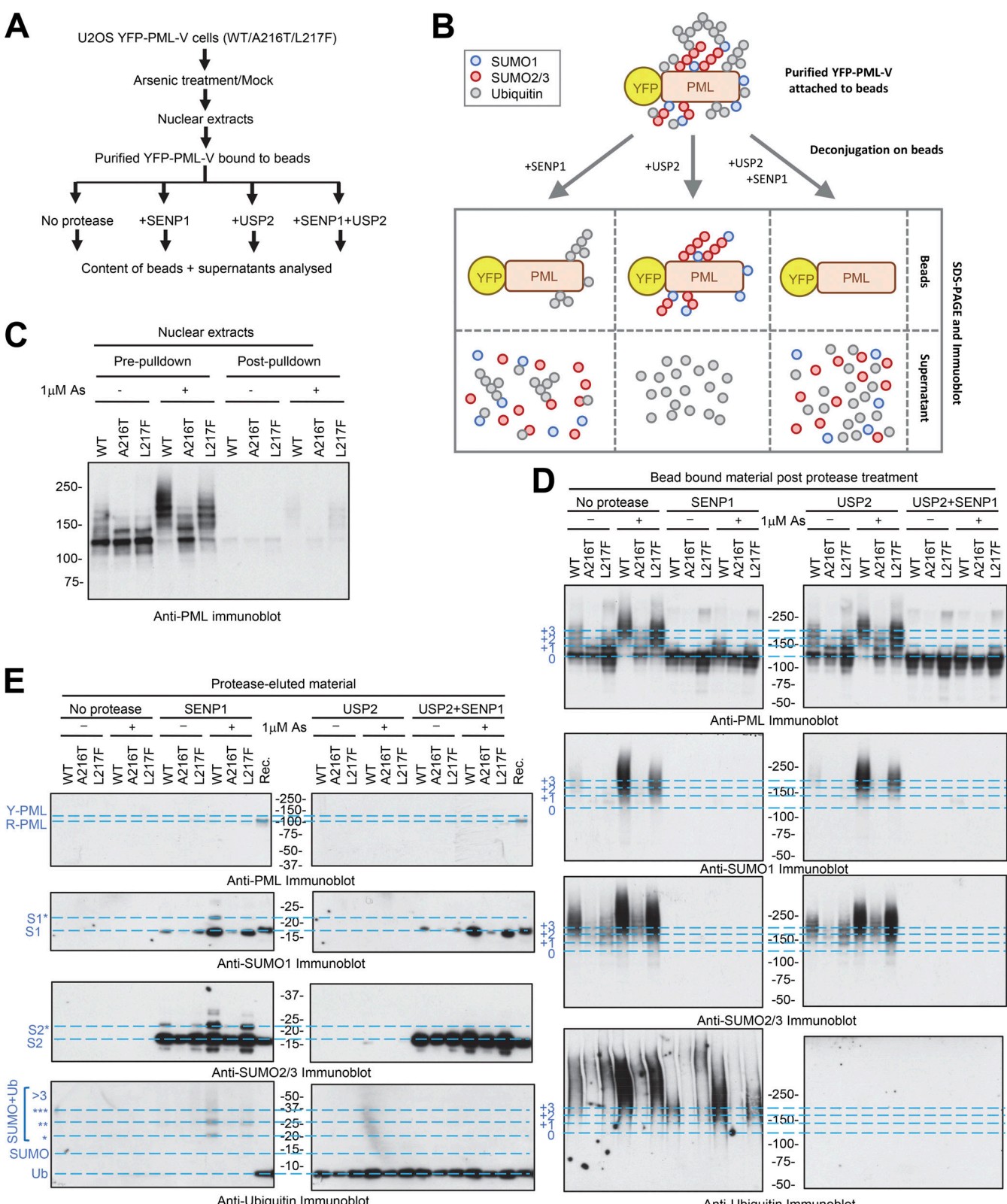

Figure 2.   **A216T and L217F mutants of PML show deficiencies to SUMO and ubiquitin conjugation in response to arsenic. (A)** Overview of the experimental design for detection of ubiquitin and SUMO conjugated to PML. **(B)** Schematic depiction of the expected contents of YFP-PML-V bound to beads and resulting supernatants after treatment with specific proteases. **(C)** Anti-PML immunoblot for nuclear extracts from the indicated cell lines before and after purification of YFP-PML-V with anti-GFP nanobody beads. **(D and E)** Analysis of the material remaining bound to anti-GFP nanobody beads (D) or eluted from the beads (E) by treatments of purified PML bodies with SUMO and ubiquitin-specific proteases. Immunoblotting used antibodies to PML, SUMO1, SUMO2/3, or ubiquitin. 2 ng standards of recombinant proteins (Rec.) were included in the eluted proteins analysis (E). The positions of recombinant PML ("R-PML") and unmodified forms of YFP-PML ("0" or "Y-PML") are indicated. Higher molecular weight modified forms of YFP-PML-V are indicated ("+1," "+2," "+3"), and ubiquitin-modified SUMO molecules are indicated with asterisks ("*," "**," "***"). Source data are available for this figure: SourceData F2.

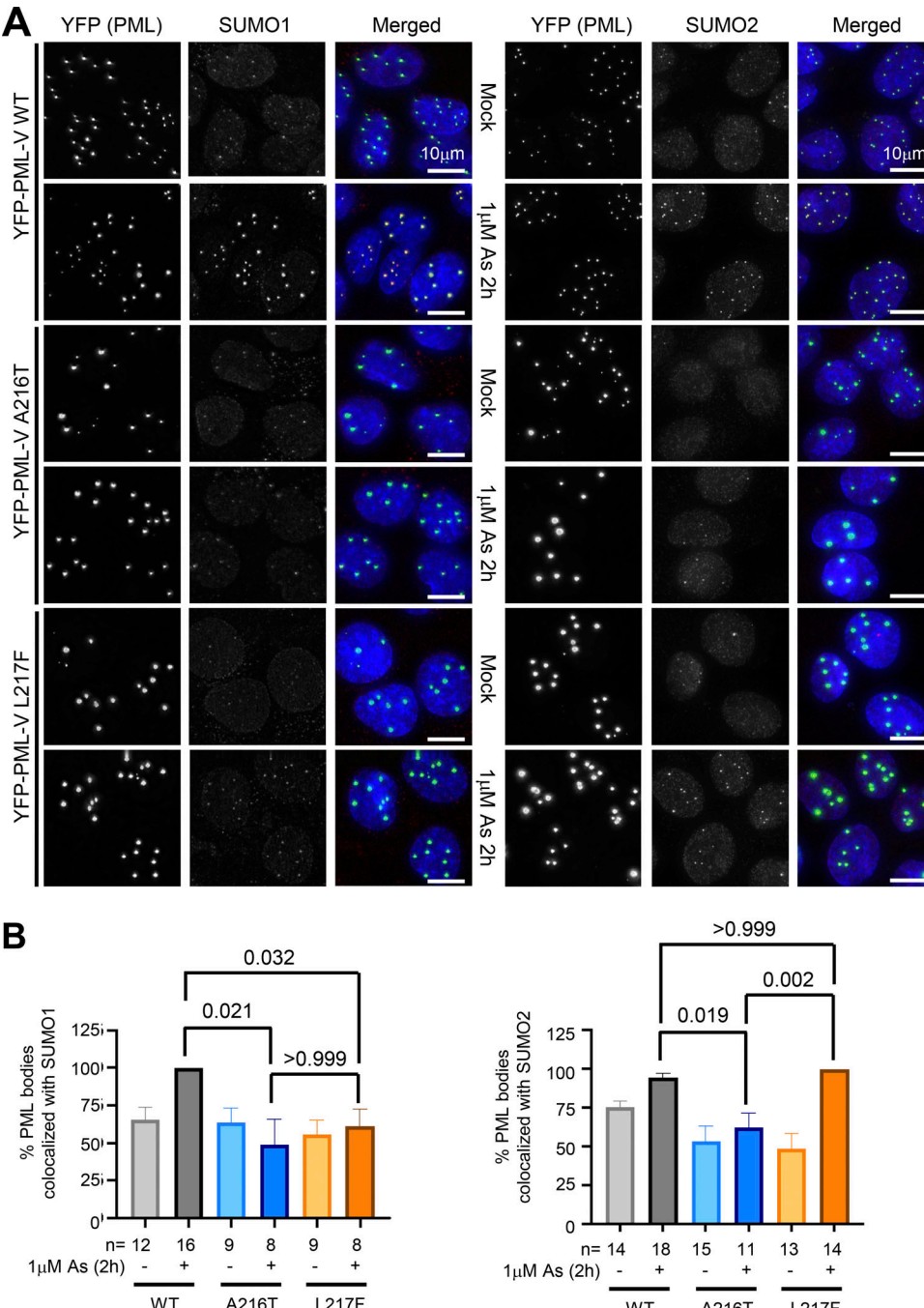

Figure 3. **A216T and L217F PML variants show altered SUMO recruitment to PML bodies upon arsenic exposure. (A)** Representative images of co-localization of SUMO1 and SUMO2/3 with WT, A216T, and L217F PML variants after 1 µM arsenic treatment for 2 h studied by fluorescence microscopy. Merge shows DAPI (blue), YFP (green), and SUMO (red). **(B)** Quantitative summary of SUMO1 and SUMO2/3 colocalization with YFP-PML. Columns are average values, and error bars are SEM. Colocalization was defined as any SUMO signal above background in the same region as a PML body. Statistical significance was assessed by comparing the arsenic-treated conditions among the three PML types using Dunn's multiple comparisons tests after Kruskal–Wallis ANOVA. The number of cells counted (*n*) is indicated in the charts. Source data are available for this figure: SourceData F3.

from WT-, A216T-, and L217F YFP-PML-V–expressing cells were again purified and the associated RNF4 was determined by immunoblotting (Fig. 4 A). RNF4 was not stably associated with nuclear bodies in the absence of arsenic, but was detected in purifications containing WT and L217F YFP-PML-V forms after arsenic treatment (Fig. 4 A). By this method, RNF4 was not detectable in any A216T YFP-PML-V purifications

(Fig. 4 A). Immunoblotting the same extracts for ubiquitin using an antibody that detects ubiquitin chains confirmed the data from Fig. 2 D; that WT YFP-PML-V was extensively ubiquitinated in response to arsenic, while ubiquitination of L217F-PML was less robust and no increase in the ubiquitination of A216T YFP-PML-V was detected (Fig. 4 A). The arsenic-induced recruitment of RNF4 to PML bodies was also

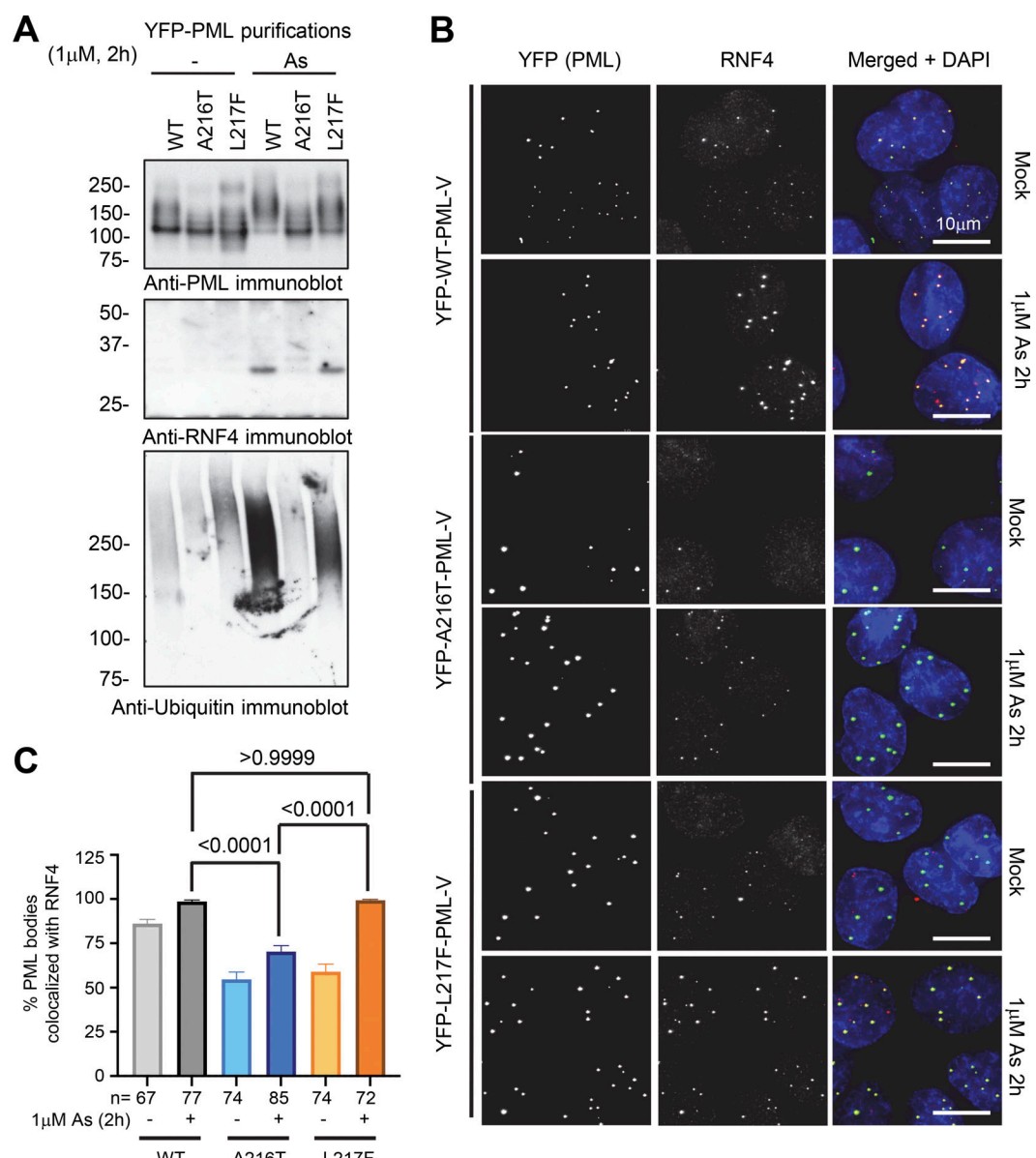

Figure 4. **RNF4 is recruited to WT and L217F-PML bodies, but not the A216T variant. (A)** Anti-PML, RNF4, and ubiquitin immunoblots from purified PML bodies containing YFP-PML-V WT, A216T, and L217F. **(B)** Representative images assessing colocalization of RNF4 with YFP-PML-V WT, A216T, and L217F after 1 μM arsenic treatment for 2 h studied by fluorescence microscopy. Merge shows DAPI (blue), YFP (green), and RNF4 (red). **(C)** Quantitation summary of the colocalization data described in B. Columns are averages with SEM error bars. Colocalization was defined as any RNF4 signal above background within a PML body. Statistical significance was assessed by comparing the arsenic-treated conditions among the three PML types using Dunn's multiple comparisons tests after Kruskal–Wallis ANOVA. The number of cells counted (n) is indicated in the chart. Source data are available for this figure: SourceData F4.

analyzed by fluorescence microscopy. This confirmed that colocalization of RNF4 with PML bodies was increased for both the WT and the L217F variant in response to arsenic, but was not observed for A216T PML (Fig. 4, B and C). These results indicate that the inability of A216T PML to be degraded is the consequence of insufficient SUMO conjugation of any type, which fails to recruit RNF4, and therefore, A216T PML is not ubiquitinated. This was not the case for L217F, which recruited RNF4 and induced ubiquitination, although this was less robust than for WT-PML and was insufficient to trigger degradation.

**In response to arsenic, L217F PML fails to recruit the p97 segregase**

It has recently been shown that the p97 segregase is required to extract ubiquitinated PML from nuclear bodies prior to degradation by the proteasome (Jaffray et al., 2023). To determine whether the mutants were compromised for p97 extraction, we analyzed WT-, A216T-, and L217F YFP-PML-V–expressing cells by fluorescence microscopy following 0-, 2-, 4-, and 6-h arsenic treatment. p97 recruitment was determined by evaluating colocalization between antibody detected p97 and PML marked by YFP fluorescence. As p97 is

a highly abundant protein, a pre-extraction procedure was necessary to release the bulk of the soluble p97 from the cells before fixation. 6-h arsenic treatment visibly increased WT YFP-PML-V association with p97, but the two mutants showed no change (Fig. 5 A). The percentage of PML bodies colocalized with p97 in WT-, A216T-, and L217F YFP-PML-V–expressing cells was determined at 0, 2, 4, and 6 h after arsenic treatment (Fig. 5 B). In untreated WT YFP-PML-V–expressing cells, about 20% of PML bodies were associated with p97, which increased to over 60% at 2, 4, and 6 h after arsenic treatment. In untreated cells expressing A216T and L217F YFP-PML-V, p97 association was <10% and this did not increase in response to arsenic (Fig. 5 B). By 6 h, the recruitment of p97 to WT-PML bodies was significantly higher than either mutant (Fig. 5 C). Importantly, total p97 levels do not differ between cell types (Fig. 5 D). Thus, although L217F YFP-PML-V recruits SUMO and RNF4 to generate ubiquitin conjugates, this signal is insufficient to engage p97, and therefore, proteasomal degradation fails.

### Quantitative proteomics analysis of PML posttranslational modifications in response to arsenic

To obtain a more quantitative and site-specific understanding of the posttranslational modifications associated with PML, a proteomics study was undertaken. Modifications including phosphorylation and ubiquitination can be identified by routine methods, but trypsin digestion of WT endogenous SUMOs leaves long C-terminal peptide adducts (Fig. S1 B, red boxes), which are challenging to identify by mass spectrometry (Hendriks et al., 2015). Thus, a strategy combining trypsin and GluC digestion was employed to give multiple, shorter SUMO C-terminal peptide fragments enabling better identification in proteomics studies (Fig. S1 B, blue boxes).

Cultures of U2OS PML$^{-/-}$ +YFP-PML-V cells were grown for the WT, A216T, and L217F PML variants, and were either treated or not with 1 μM arsenic for 2 h (Fig. S1 C). Four experimental replicates for each condition were prepared, and YFP-PML-V cells purified by the same method as described above. Purifications were fractionated by SDS-PAGE, and the section of the gel containing YFP-PML-V was excised for analysis (Fig. S1 D). Anti-PML immunoblot of a fraction of the inputs showed the expected YFP-PML-V patterns (Fig. S1 E). After peptide analysis by LC-MS/MS, multiple rounds of data processing were undertaken to maximize numbers of modifications (see Materials and methods), which ultimately yielded 21 sites of phosphorylation, 14 of SUMOylation, and 11 of ubiquitination in PML-V (Fig. S1 F and Data S1). Principal component analysis using intensity data for all modified and unmodified peptides from YFP-PML, SUMO1, SUMO2, SUMO3, and ubiquitin shows replicates cluster by experimental condition and are clearly separated from one another (Fig. 6 A), indicating consistency among replicates and differences between conditions.

PML phosphorylation was largely localized to the N and C termini, and in all cases, identified phosphorylations were proximal to prolines (Fig. S2 A), suggesting the process is largely proline-directed. The overall phosphorylation status of each PML type was compared using the sum of all phosphopeptide

intensities relative to untreated WT-PML (Fig. S2 B), which suggests there are no large-scale differences among the PML types either before or after arsenic treatment. Broken down by site, a broadly similar pattern is seen across all PML variants either in the presence or in the absence of arsenic (Fig. S2 C). Thus, while it cannot be excluded that differential phosphorylation is relevant to the mutant PML phenotypes, these data do not suggest that differences in phosphorylation status play an important role.

### Site-specific differences in SUMO modification to PML mutants

To gain a broad overview of the relative SUMO and ubiquitin modification status of the PML variants, the total intensity of the unmodified peptides from SUMO1, SUMO2/3, and ubiquitin was used as a proxy for their overall protein abundance in YFP-PML purifications (Fig. 6 B). This showed a consistent pattern to the immunoblot experiments described above (see Fig. 2). Namely, arsenic treatment has little effect on the SUMO or ubiquitin modification status of the A216T mutant, but triggers increased conjugation by SUMO1, SUMO2/3, and ubiquitin for WT and L217F PML, with the mutant showing lesser accumulation of SUMO1 and ubiquitin than WT-PML. Also consistent with the immunoblot data is the finding that the relative increase in SUMO1 conjugation to WT-PML is much greater (fivefold) than for SUMO2/3 (1.8-fold) (Fig. 6 B).

Ten SUMO1 modification sites in PML provided data good enough for comparisons among conditions (Fig. 6 C, Fig. S2 D, and Data S1). Relative PML site occupancy by SUMO1 was estimated by signal intensity of peptides indicative of specific modifications (Fig. 6 D). While peptide signal intensity is not directly proportional to abundance for peptides of different sequence, for branched peptides which share a large proportion of their sequence (from the common SUMO adduct), signal intensity is a better approximation of abundance. Based on this assumption, the major acceptor of SUMO1 on all PML types under all conditions is K65 in the RING domain (Fig. 6 D). SUMO1 conjugation at K65 also increases with arsenic treatment for all PML types, although the occupancy for the A216T mutant is relatively low compared with WT and L217F (Fig. 6 D). K65 has already been identified using mutational analysis as one of three major SUMO acceptors (Kamitani et al., 1998). Peptides diagnostic of SUMO1 modification to another of these three sites, K490 in the NLS domain, also gave a strong intensity in our samples, although this is most intense in the WT-PML samples after arsenic treatment (Fig. 6 D). The third major SUMO acceptor according to mutational analysis, K160 in B-Box1, gave a relatively low SUMO1-modified peptide intensity signal in all PML variant purifications (Fig. 6 D and Fig. S2 E), suggesting SUMO1 occupancy here is not high. Another striking difference is the large increase in WT-PML for SUMO1 modification at K380 in the region between the coiled-coil and the NLS, which appears to be blunted for both mutants (Fig. 6 E and Fig. S2 E). All other SUMO1 modified lysines in this region and the NLS itself (K400, K401, K426, K478, K487, K490, and K497) show strong induction with arsenic for WT-PML that is absent in A216T PML and muted in L217F (Fig. S2 E). Importantly, the only

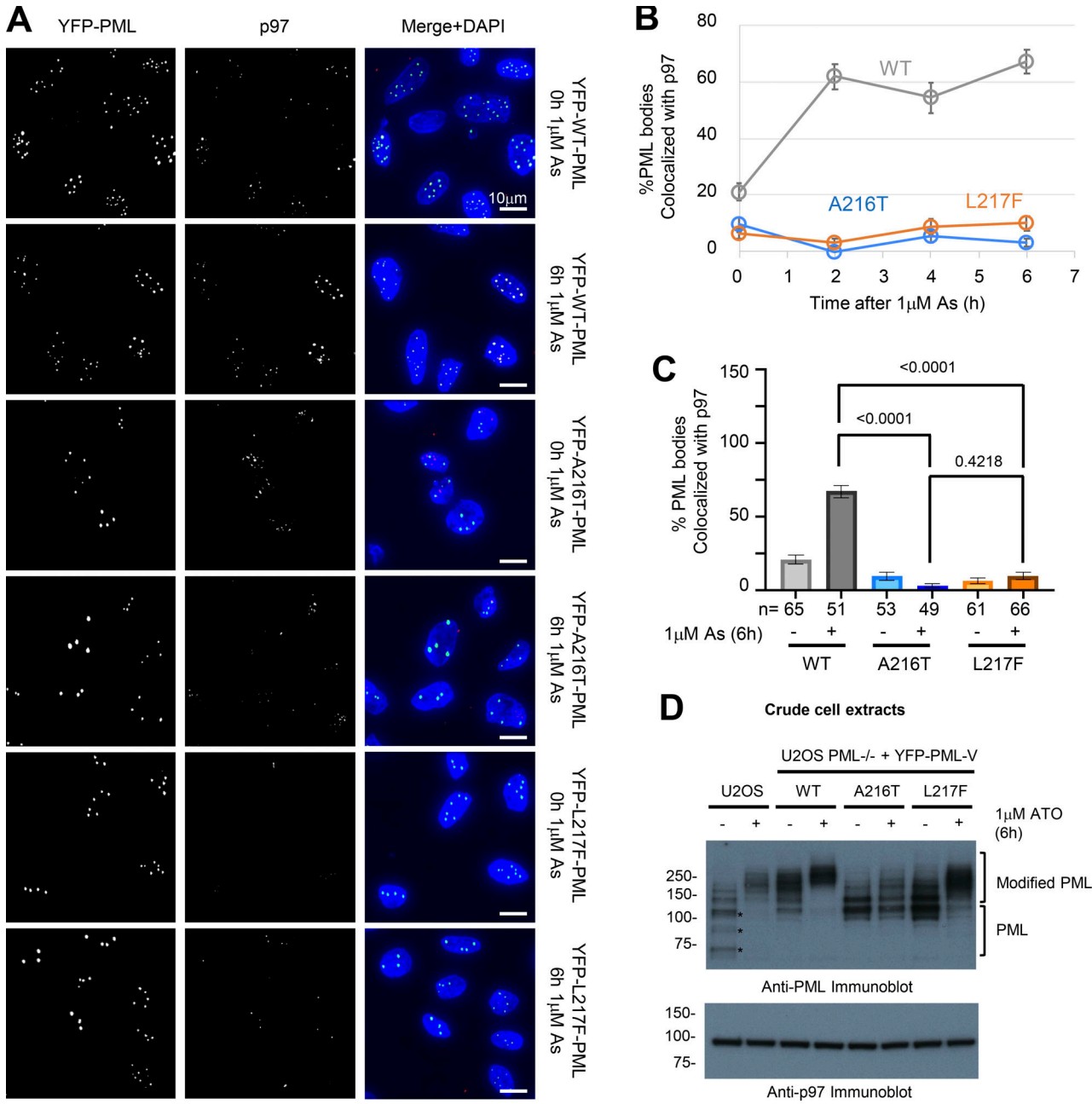

Figure 5. **Neither A216 nor L217F mutants of PML recruit p97 to nuclear bodies upon arsenic treatment. (A)** Representative fluorescence microscopy images for 1 μM arsenic-treated YFP-PML-V cells at 0 and 6 h, showing YFP fluorescence and p97 immunofluorescence. **(B)** Quantitative summary of p97 colocalization with PML bodies in the indicated cell lines during 0-, 2-, 4-, and 6-h arsenic exposure. Markers show average values, and error bars are SEM. **(C)** Statistical summary of the comparison between 0- and 6-h arsenic exposure for YFP-PML-p97 colocalization. Columns represent average % colocalization per cell, and error bars are SEM. Cell count (*n*) is shown in brackets below each column. Statistical significance was assessed by comparing the arsenic-treated conditions among the three PML types using Dunn's multiple comparisons tests after Kruskal–Wallis ANOVA. **(D)** Anti-PML and anti-p97 immunoblots of crude cell extracts taken from the indicated cell lines either treated or not with 1 μM arsenic for 6 h; WT U2OS (U2OS) or PML$^{-/-}$ U2OS expressing the YFP-PML-V variants (WT, A216T, or L217F). *Note multiple modified and unmodified endogenous PML isoforms are difficult to distinguish. Source data are available for this figure: SourceData F5.

site with significantly higher SUMO1 occupancy for any mutant compared with WT-PML is K160, which is modestly but significantly more occupied for L217F-PML during arsenic treatment (Fig. S2, E and F).

14 sites of SUMO2/3 conjugation to PML were identified, including all 10 SUMO1 sites along with four additional sites at

K226, K394, K460, and K476 (Fig. 6 C). Interestingly, the four most intense SUMO2/3 acceptors, K65, K160, K380, and K490, are all within SUMO conjugation consensus or reverse consensus motifs (Fig. S1 G). As with SUMO1, K65 is a major acceptor for SUMO2/3 conjugation and shows a similar pattern of arsenic induction (compare Fig. 6 D with Fig. 6 E and Fig. S2 E, left and

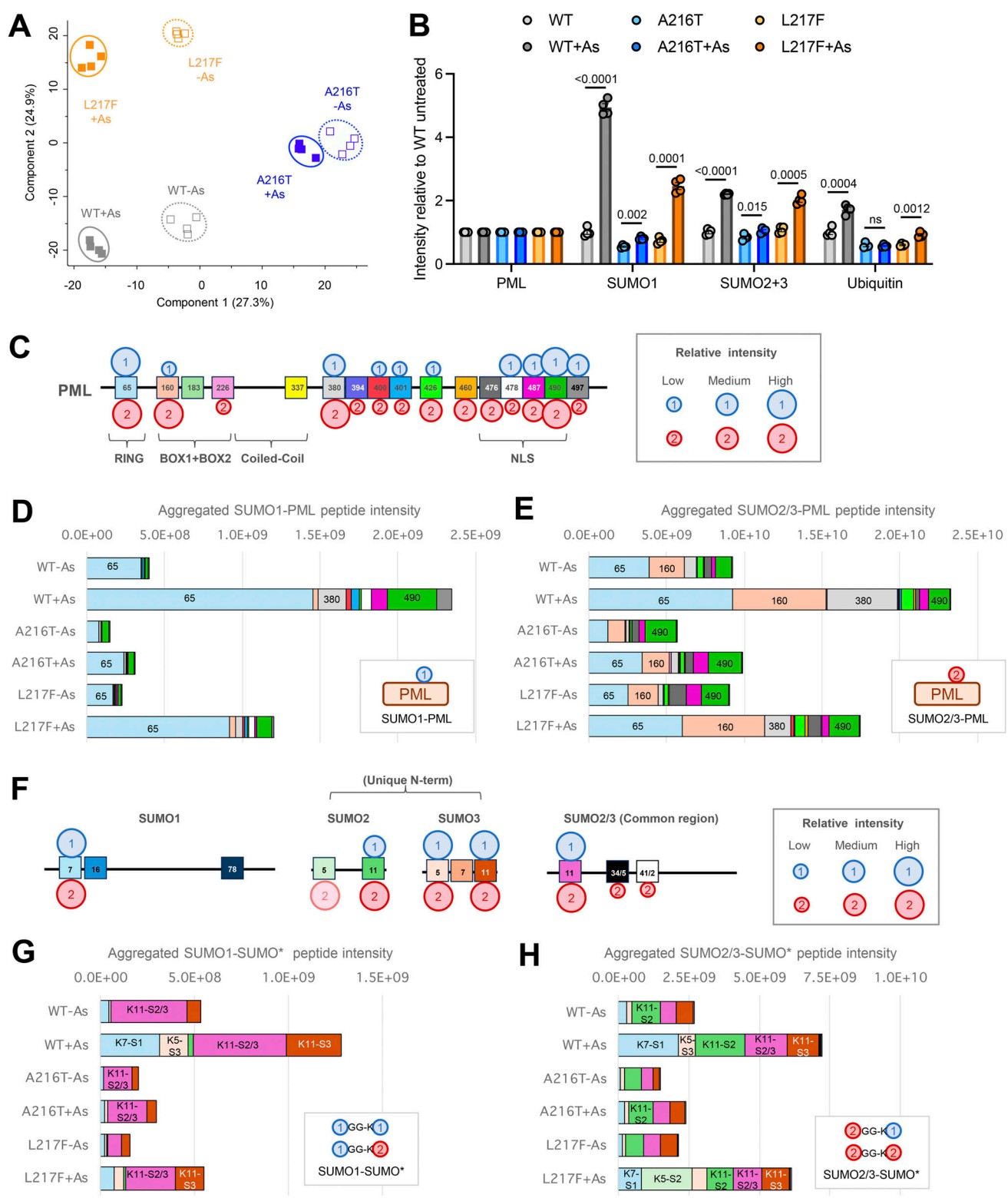

Figure 6.   **YFP-PML WT, A216T, and L217F show differential SUMOylation at the site level.** Data derived from the proteomics experiment detailed in Fig. S1. **(A)** Principal component analysis based on all modified and unmodified peptides detected from SUMO1, SUMO2, SUMO3, ubiquitin, and PML. **(B)** Relative protein intensity for SUMO1, SUMO2+3, ubiquitin, and PML in the purifications. Protein intensities are the sum of all unmodified peptide intensities and are a proxy for protein abundance. SUMO2+3 is the sum of the intensities all peptides derived from both paralogs. Number of replicates *n* = 4, and significance was determined by unpaired *t* tests with Welch's correction where necessary. **(C)** Schematic presentation of PML with sites of lysine modifications indicated relative to PML domains. SUMO1 and SUMO2/3 modifications are shown as circles, with size approximating to peptide intensity. **(D)** Aggregated SUMO1-PML peptide intensity data for each branched peptide identified in each YFP-PML purification. **(E)** Aggregated SUMO2/3-PML peptide intensity data for each branched peptide identified in each YFP-PML purification. **(F–H)** Schematic presentation of SUMO1, SUMO2, and SUMO3 sequences with sites of lysine

modifications indicated. SUMO1 and SUMO2/3 sites are shown by circles, with size approximating to peptide intensity. Paler circles were not detected in WT-PML purifications. **(G and H)** Aggregated peptide intensity data for each branched peptide for SUMO1 (G) or SUMO2/3 (H) conjugated to an acceptor SUMO molecule.

right). Also consistent with SUMO1, SUMO2/3 conjugation at K380 is strongly induced for WT-PML, modestly induced for L217F-PML, and apparently absent for the A216T mutant (Fig. 6 E and Fig. S2 E). The most striking contrast with the SUMO1 data is the much higher relative intensity of the SUMO2/3-modified K160 peptide (compare Fig. 6 D with Fig. 6 E), supporting the idea K160 may be a SUMO2/3 preferential site, as previously proposed (Lallemand-Breitenbach et al., 2008). An additional difference with SUMO1 is many SUMO2/3 conjugation sites show higher occupancy for the mutants than WT (Fig. S2 F boxed P values). For example, K226 has higher SUMO2/3 occupancy in A216T-PML than either WT or L217F-PML (Fig. S2 E), which may be a consequence of structural differences close to the sites of mutation. Furthermore, K490 SUMO2/3 occupancy is modestly but significantly lower for WT-PML than either mutant, a pattern consistent for most NLS lysines (Fig. S2, E and F). This contrasts with SUMO1 modification in the NLS, which shows much higher conjugation in WT-PML (compare Fig. S2 E, "NLS," left and right). This reciprocal pattern between SUMO1 and SUMO2/3 conjugation in NLS lysines implies competition between SUMO paralogs, and that upon arsenic treatment, WT-PML accumulates more SUMO1 and less SUMO2/3 than either mutant. Notably, K65, K160, and K490 are the only three sites for which the unmodified counterpart peptides showed large and significant differences among samples (Fig. S3, A–E), supporting the idea these may be high occupancy sites where SUMO1 and SUMO2/3 compete for attachment. Other sites, while detectably conjugated by SUMOs, may have much lower occupancy.

## SUMO polymer formation on WT-PML and the A216T and L217F mutants

SUMO polymers are key signals in the PML degradation pathway. To assess the nature of the SUMO polymers attached to each PML type, peptides indicative of SUMO-SUMO linkages were quantified in the proteomics data (Data S1). In total, 8 SUMO-SUMO branched peptides were identified (Fig. 6 F). Data were divided into two categories: SUMO-1 conjugating to any SUMO type, or SUMO-2/3 conjugating to any SUMO type, and aggregated intensity data calculated for each (Fig. 6, G and H). This shows the abundance of SUMO1 and SUMO2/3 polymers broadly mirror SUMO1 and SUMO2/3 conjugation to PML itself (compare Fig. 6, D and E with Fig. 6, G and H). Specifically, A216T-PML is associated with lower levels of SUMO polymers both before and after arsenic exposure than either WT or L217F-PML. L217F PML shows a similar scale of SUMO2/3 polymer formation to WT-PML (Fig. 6 H), but SUMO1 polymers are approximately half WT levels (Fig. 6 G). No SUMO-SUMO linkage is more abundant in purifications from either mutant than for the WT (Fig. S4, A and B). These data support the hypothesis that WT-PML is associated with more extensive SUMO polymers

than either mutant and that it is incorporation of the SUMO1 paralog into these polymers that shows the most striking difference between WT and the mutant.

## Differential ubiquitination associated with WT-PML and A216T and L217F mutants

Ubiquitin can be covalently associated with PML in three ways: conjugated directly to PML, conjugated indirectly via SUMO, or by participating in polyubiquitin chains attached to either. 11 ubiquitination sites in PML were identified across all samples (Fig. 7 A). Comparisons among PML variants for total ubiquitin-PML peptide intensity show direct ubiquitination of A216T-PML is relatively low, while WT and L217F types show similar total levels of direct ubiquitination after arsenic treatment (Fig. 7 B). This does not closely match the pattern of overall ubiquitin amounts associated with each purification (compare with Fig. 6 B, ubiquitin), suggesting a large fraction of total ubiquitin associated with WT-PML is not directly conjugated to the PML protein. Although 11 ubiquitination sites were identified across all samples, only seven provided intensity information for WT-PML: K183, K337, K380, K394, K400, K426, and K476. Interestingly, none of the three major SUMO acceptors (K65, K160, and K490) were detectably ubiquitinated on WT-PML, but K160 and K490 were ubiquitinated for both mutants (Fig. S5, A–C) implying that the mutations either directly or indirectly affect the lysines in PML available for ubiquitination. For WT-PML, the bulk of the ubiquitinated peptide intensity comes from three residues within the domain between the coiled-coil and NLS: K337, K380, and K394 (Fig. 7 B). According to our data, K337 and K394 are poor SUMO acceptors (Fig. 6, D and E), suggesting the SUMO and ubiquitin conjugation (or deconjugation) systems have preference for a different subset of lysines in PML.

Considering the ubiquitination of SUMO molecules (Fig. 7 C), there is an unexpected finding that SUMO ubiquitination in the absence of arsenic is lower for WT-PML than either mutant (Fig. 7 D). However, during arsenic exposure A216T-PML does not experience a large increase in SUMO ubiquitination, while the L217F mutant accumulated more ubiquitin conjugated to SUMOs than WT-PML (Fig. 7 D). Specifically, 4 of the 5 ubiquitination sites in SUMO 1 and SUMO2/3 were more extensively ubiquitinated in L217F PML purifications than WT (Fig. S6, A and B), suggesting that while L217F is a loss-of-function mutant in the context of overall degradation in response to arsenic, it has higher than WT levels of SUMO ubiquitination.

The third type of ubiquitin linkage associated with PML, Ub-Ub conjugates, includes the signals recognized by p97 and the degradation machinery. Four Ub-Ub linkage-specific peptides were detected and quantified (Fig. 7 E). None of these showed an arsenic-induced change for either mutant (Fig. 7 F). WT-PML does respond with significantly increased polyubiquitination via K6, K11, and K48, although the absolute change is only modest

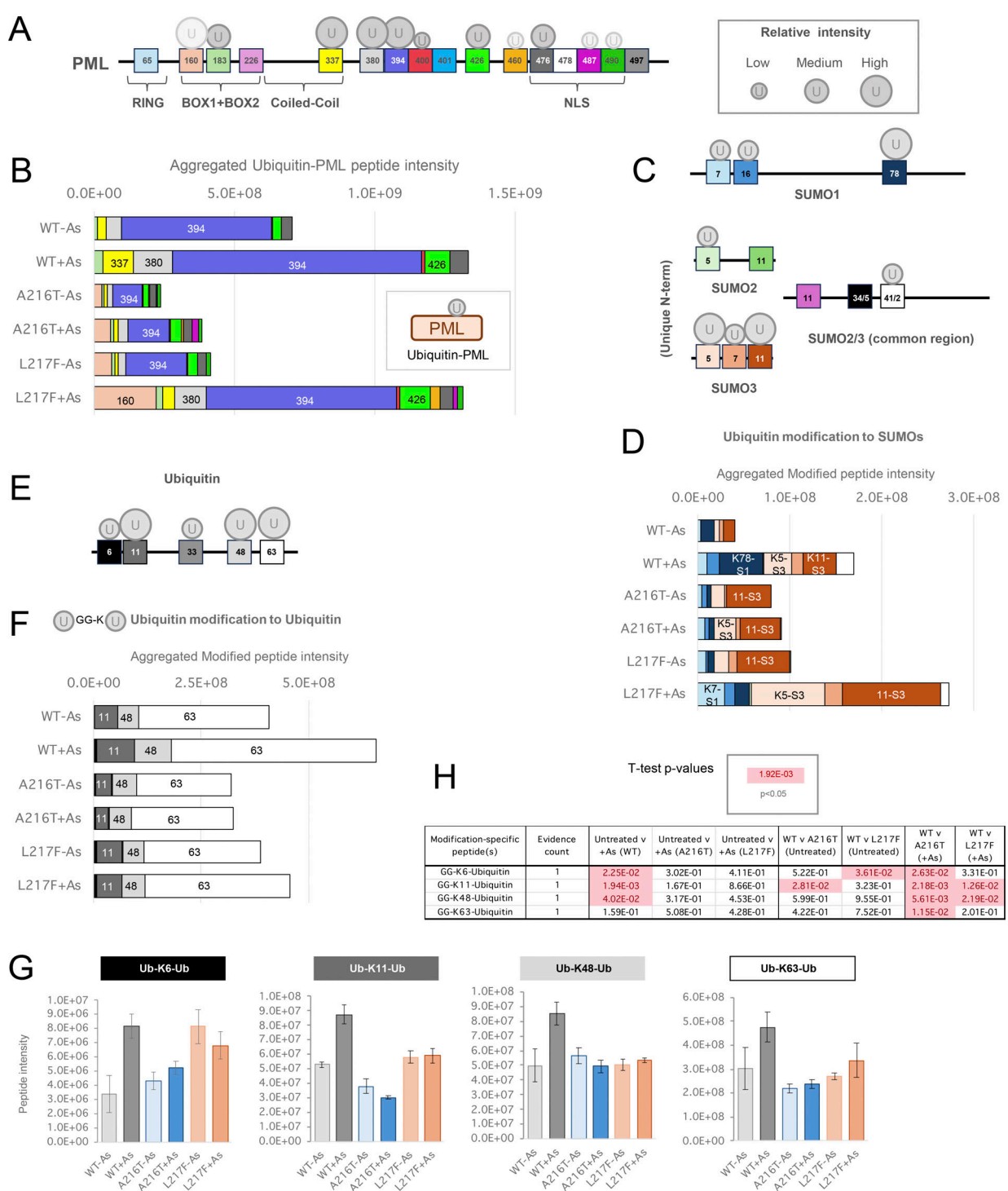

Figure 7. **YFP-PML WT, A216T, and L217F show differential ubiquitination at the site level. (A)** Schematic presentation of PML sequence with sites of lysine modifications indicated. Ubiquitination sites are shown by circles, with size approximating to peptide intensity. Paler circles were not detected in WT-PML purifications. **(B)** Aggregated ubiquitin-PML peptide intensity data for each GlyGly-K peptide identified in each YFP-PML purification. **(C)** Schematic presentation of SUMO sequences with sites of lysine ubiquitination indicated as described for A. **(D)** Aggregated ubiquitin-SUMO peptide intensity data for each GlyGly-K peptide identified in experimental condition. **(E)** Schematic presentation of ubiquitin sequence with ubiquitination sites of ubiquitination shown as described for A. **(F)** Aggregated ubiquitin–ubiquitin peptide intensity data for each GlyGly-K peptide identified in each YFP-PML purification. **(G)** Charts of average intensity (columns) and SEM (bars) for the GlyGly-K peptides diagnostic of ubiquitin–ubiquitin modifications at the indicated sites (number of purifications, n = 4). **(H)** t test summary for the pairwise comparisons indicated for each ubiquitin–ubiquitin modification. Welch's correction was applied for heteroskedastic data. Red entries are P < 0.05. Number of different peptides used for each site is indicated as evidence count.

(Fig. 7, G and H). This is consistent with previous work, which showed that ubiquitinated PML is rapidly detected by p97, leading to PML turnover, which makes detection of poly-ubiquitin signals associated with PML difficult in the absence of p97 inhibitors (Jaffray et al., 2023). These results support a hypothesis whereby only WT-PML accrues the polyubiquitin signals necessary for p97 recruitment and rapid extraction from PML bodies, prior to degradation by the proteasome.

**SUMO1 is essential for arsenic-induced PML degradation**
To confirm that SUMO modification is required for arsenic-induced degradation of YFP-PML-V in our PML⁻/⁻ U2OS cells, they were pretreated with the highly specific and potent SUMO E1 inhibitor ML792 (He et al., 2017), before exposure to arsenic and YFP fluorescence monitored by live-cell imaging. ML792 will block conjugation of all SUMO paralogs. Images taken at 0 and 16 h after arsenic administration confirm that in the absence of ML792, PML was degraded and PML bodies are undetectable (Fig. 8 A, upper left). In the presence of ML792, PML is not degraded and PML bodies are still evident after 16-h exposure to arsenic (Fig. 8 A, upper right). Quantitation of the real-time imaging data confirms that ML792 completely blocks arsenic-induced degradation of PML (Fig. 8 A, lower). To investigate the specific role of SUMO1 in arsenic-induced PML degradation, a U2OS cell line lacking SUMO1 (SUMO1⁻/⁻ U2OS) was generated by CRISPR/Cas9 genome editing. Immunofluorescence confirmed the loss of SUMO1 expression in these cells (Fig. 8 B). Anti-PML immunoblot analysis of crude cell extracts of cells exposed to 1 µM arsenic over 24 h showed that in cells lacking SUMO1, there is a molecular weight shift in PML in a manner similar to SUMO1⁺/⁺ cells (Fig. 8 C), but the modified forms of PML appear to be more resistant to degradation. These persist up to 24 h after the addition of arsenic to the medium, when PML from WT cells is almost completely degraded (Fig. 8 C). Furthermore, after these extended periods of arsenic exposure, unconjugated PML begins to re-accumulate in SUMO1⁻/⁻ cells (Fig. 8 D), almost returning the cellular pool of PML to the initial state (compare 0- and 24-h lanes in Fig. 8 D). This confirms SUMO1 is required for efficient PML degradation, and insufficient conjugation of the L217F PML mutant by SUMO1 may explain its resistance to arsenic-induced degradation.

**L217F PML fails to recruit TOPORS in response to arsenic**
In addition to RNF4, two further STUbLs have been implicated in PML turnover: RNF111 (Arkadia) (Erker et al., 2013) and TOPORS (Liu et al., 2024). To investigate any potential links between RNF111 and TOPORS with these PML mutants, the YFP-PML proteomics data were reanalyzed for copurified proteins, this time including peptides derived from all regions of the gel (Fig. S1 D) (Data S1). Both RNF111 and TOPORS were detected in the YFP-PML purifications from all samples, and their abundance was highest in WT-PML purifications after arsenic treatment (Fig. 9 A). While arsenic treatment resulted in a fivefold increase in the amount of TOPORS recruited to WT-PML bodies, there was only a modest increase in RNF111, which is less than doubled when cells were exposed to arsenic (Fig. 9 A). For A216T-PML, TOPORS levels were unaffected by arsenic, and

while TOPORS levels did increase with the L217F mutant, even after arsenic, amounts were only equivalent to the untreated WT-PML levels (Fig. 9 A). Notably, the relative abundance of TOPORS across all samples closely correlated with SUMO1 (Fig. 9 A). To determine the requirement for RNF111, TOPORS, and RNF4 in the arsenic-induced degradation of YFP-PML-V WT in U2OS PML⁻/⁻ cells, siRNA was used to deplete all three (Fig. 9, B and C), followed by live-cell imaging of YFP-PML WT during arsenic exposure. This showed that RNF111 ablation appeared to have no significant effect on PML degradation, while RNF4 and TOPORS individually did inhibit PML degradation (Fig. 9, D–F). Cells lacking both RNF4 and TOPORS were almost unresponsive to arsenic (Fig. 9, D–F). The requirement for TOPORS in PML degradation was further confirmed by CRISPR/Cas9 knockout of the TOPORS gene (Fig. S7 A), which showed inhibited YFP-PML turnover when TOPORS guides were cotransfected with Cas9 (Fig. S7, B and C).

These results raise the possibility that it is SUMO1 that directs ubiquitin modification by TOPORS. To test this directly, we generated substrates containing multiple copies of SUMO1 (4xSUMO1) or SUMO2 (4xSUMO2) and carried out in vitro ubiquitination assays with a bacterially expressed and purified fragment of TOPORS (2-574) (Fig. 9 G). Using the E2 UbcH5a, TOPORS monoubiquitinated 4xSUMO1 but showed little activity toward 4xSUMO2 as a substrate (Fig. 9 H, upper panels). The extent of TOPORS autoubiquitination was unaffected by substrate type (Fig. S7 D, upper panels). To establish whether TOPORS was functioning as a STUbL, the 3 SIMs in this fragment were individually and collectively mutated (Fig. 9 G). While mutation of either SIM1 or SIM3 in TOPORS had little impact on ubiquitination, mutation of SIM2 substantially reduced 4xSUMO1 ubiquitination (Fig. 9 H, upper panels), confirming TOPORS is acting as a STUbL. For comparison, previously described RNF4 SIM mutants (Tatham et al., 2008; Xu et al., 2014) were used. RNF4 ubiquitinated both 4xSUMO1 and 4xSUMO2 to a similar extent, and the SIM mutations reduced ubiquitination activity as shown previously (Tatham et al., 2008; Xu et al., 2014) (Fig. 9 H, lower panels). Thus, it appears that in response to arsenic, L217F PML is modified by SUMO2/3 and recruits RNF4, but is defective for SUMO1 modification and fails to recruit TOPORS. Consequently, the ubiquitin signal is not sufficient to recruit p97 and L217F is not degraded.

## Discussion

The overarching cellular mechanisms governing arsenic-induced degradation of PML and the oncogenic fusion PML-RARA have been broadly understood for over a decade; arsenic triggers the SUMOylation of PML, which targets it for ubiquitination, and degradation by the proteasome. However, it is still unclear precisely which molecular signals and effector proteins are required at each step. Initial mutational studies of PML SUMOylation identified three major acceptors at lysines 65 (in the RING domain), 160 (in B-Box1), and 490 (in the NLS) (Kamitani et al., 1998). Early proteomics data expanded upon this to include lysines 380, 400, and 497 (Galisson et al., 2011).

**Figure 8. SUMO1 is required for efficient arsenic-induced PML degradation. (A)** U2OS PML⁻/⁻ + YFP-PML-V WT cells were exposed to 0.4 µM ML792 for 2 h prior to the addition or not of 1 µM arsenic, followed by live-cell imaging over 18 h. Representative fluorescence microscopy images are shown (upper panels), and relative YFP-PML-V body intensity is normalized to time point 0 h for each frame independently and shown graphically (lower panel). Solid lines are average values, and shaded areas represent one SEM (n = 9 fields of view). **(B)** Immunofluorescence analysis of WT U2OS (SUMO1⁺/⁺) and SUMO1⁻/⁻ cells exposed to 1 µM arsenic for 2 h, fixed, and double-stained with DAPI and anti-SUMO1 (Alexa Fluor 488). **(C)** SUMO1⁺/⁺ (wt) and SUMO1⁻/⁻ (KO) U2OS cells were exposed to 1 µM arsenic for the indicated periods before lysis and analysis by immunoblot for PML. **(D)** PML immunoblot from crude extracts of SUMO1⁻/⁻ U2OS cells after exposure to 1 µM arsenic for the indicated times. * Nonspecific background band. Source data are available for this figure: SourceData F8.

More recent large-scale SUMO site proteomics studies have increased this further, with the single largest SUMO study to date (Hendriks et al., 2017) describing a total of 15 SUMOylation sites on PML. Ubiquitin and the three SUMO paralogs all conjugate to lysine residues within PML, as well as each other, giving scope for a highly complex PML–SUMO–ubiquitin axis with the potential to generate dozens of different posttranslational signals. Therefore, a significant challenge is determining which signals are relevant to the process of PML degradation in response to arsenic. Detailed characterization of mutant forms of PML and PML-RARA found in APL patients refractory to arsenic treatment offers an opportunity to expand our understanding of arsenic-induced PML degradation by exploring the underlying causes of their loss of function.

Using a combination of high-content imaging, time-lapse microscopy, and quantitative proteomics analysis to study two common patient-derived PML mutants, A216T and L217F, we have further refined our understanding of this pathway. In untreated cells, the two mutants of PML formed atypical nuclear bodies, fewer in number and larger in size than WT bodies, and both mutants were stable when cells were exposed to 1 µM arsenic for 24 h. Surprisingly, during arsenic treatment the modification status of each was different, and both differed from WT-PML. As expected, after 2-h exposure to 1 µM arsenic, WT-PML shows increased modification by SUMO1, SUMO2/3, and ubiquitin, and by 24 h is almost completely degraded. In contrast, the same arsenic treatment had no significant effect on A216T PML, which showed little increase in either SUMOylation or ubiquitination and was stable even after 24 h. Thus, the stability of A216T PML is explained by compromised conjugation of all SUMO paralogs. Surprisingly, arsenic did trigger both SUMO and ubiquitin conjugation to L217F PML, and yet after 24 h, it too was stable. Proteomics analysis revealed that the site-specific nature of the ubiquitin conjugation was also different

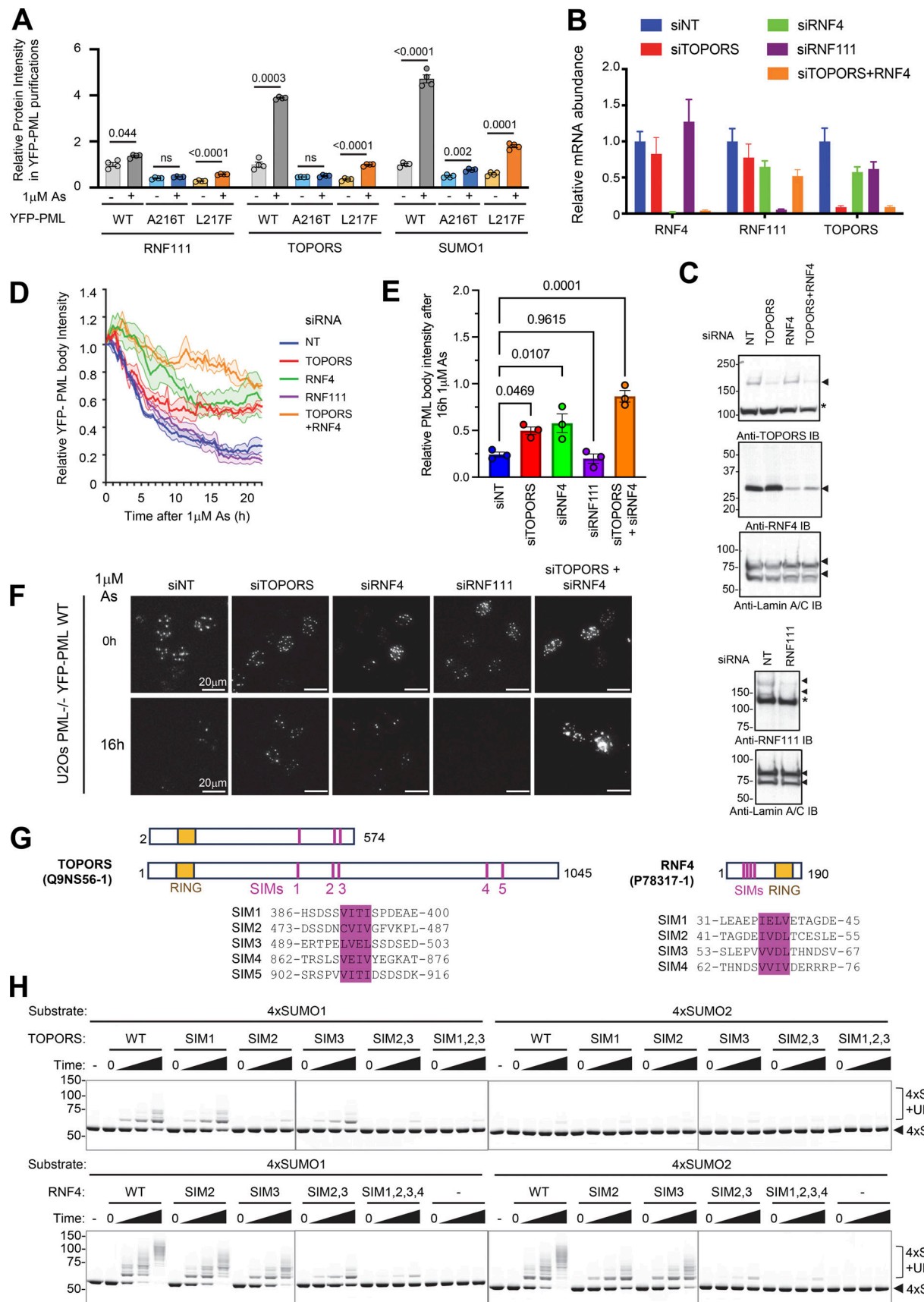

Figure 9. **TOPORS is a SUMO1-specific ubiquitin E3 ligase required for efficient PML degradation. (A)** Total peptide (protein) signal intensity for TOPORS, RNF111, and SUMO1 associated with WT, A216T, and L217F PML bodies before or after exposure for 2 h with 1 μM arsenic. SUMO1 is included for

comparison. Columns represent average, and error bars are SEM. Number of purifications, $n = 4$. P values are derived from Brown–Forsythe and Welch ANOVA tests with Dunnett's T3 multiple comparisons test. Experimental details are shown in Fig. S1. **(B)** Relative mRNA abundance normalized to TATA binding protein (TBP) expression for RNF4, RNF111, and TOPORS, in U2OS PML$^{-/-}$ +YFP-PML-V WT cells transfected with the indicated siRNAs. **(C)** Immunoblots for RNF111, TOPORS, and RNF4 from crude extracts taken from U2OS PML$^{-/-}$ +YFP-PML-V WT cells transfected with the indicated siRNAs. **(D)** Summary of live-cell microscopy analysis of PML body intensity for U2OS PML$^{-/-}$ + YFP-PML-V cells transfected with either non-targeting siRNA (siNT) or the indicated siRNAs for 48 h and exposed to 1 μM arsenic for 24 h; values are average (solid lines) with SEMs (shaded areas) of PML body intensity relative to t = 0. $n = 3$ fields of view containing multiple cells. **(E)** Summary of the relative YFP-PML body intensity data shown in D for t = 16 h. P values are derived from Brown–Forsythe and Welch ANOVA tests with an unpaired $t$ test using Welch's correction. **(F)** Representative cell images from live-cell microscopy summarized in D (scale bars are 20 μm). **(G)** Schematic depiction of the primary sequences for TOPORS and RNF4 with RING and SIMs (SIM) indicated. **(H)** Fluorescent scans of gels fractionating the products of in vitro ubiquitin conjugation reactions using Alexa Fluor 647–labeled linear fusions of 4xSUMO1 and 4xSUMO2 as substrates. E3 ligase activities of lipoyl domain–tagged WT TOPORS (2–574) ("TOPORS") and untagged WT RNF4 (full-length) are compared with the indicated SIM mutants. Assays were either stopped prior to the addition of ATP (0) or incubated for 5, 10, or 30 min after the addition of ATP. Control samples showing Alexa Fluor 647-4xSUMOs and lacking all ubiquitin conjugation machinery (−) are also included. Unmodified 4xSUMOs (4xS) and ubiquitinated 4xSUMOs (4xS+Ub) are indicated. Gels were also Coomassie-stained (Fig. S7 E). Source data are available for this figure: SourceData F9.

from WT-PML. Compared with WT-PML, the ubiquitin conjugation associated with L217F-PML appeared to be skewed toward modification of associated SUMO molecules and away from Ub-Ub polymers (Fig. 10 A). Indeed, none of the Ub-Ub linkage–specific peptides detected in our analysis increased significantly for either mutant during arsenic treatment (Fig. 7, G and H), while K6, K11, and K48 Ub-Ub linkages showed modest but significant increase in WT-PML. This is consistent with work showing K11 and K48 linkages are implicated in p97 binding (Locke et al., 2014; Meyer and Rape, 2014; Yau et al., 2017), which is the next step of the pathway (Jaffray et al., 2023). Therefore, we propose that while ubiquitination of L217F-PML does increase after arsenic exposure, this does not lead to the generation of an appropriate polyubiquitin signal. It is important to note that the relatively small increase in K11- and K48-linked Ub-Ub polymers on WT-PML after arsenic treatment (Fig. 7 G) is likely to be because the PML molecules harboring these signals are rapidly recognized and turned over (Jaffray et al., 2023). Considering the rapid SUMOylation response upon arsenic exposure (Fig. 1 F), this suggests that the generation of appropriate polyubiquitin chains is the rate-limiting step of the PML degradation pathway.

Why does L217F PML not accrue the polyubiquitin signals required to promote degradation? The answer appears to be, at least in part, due to an upstream imbalance in SUMO paralog–specific modification affecting ubiquitin-ligase recruitment. While WT and L217F forms of PML accumulate comparable amounts of SUMO2/3 after exposure to arsenic (Fig. 6, B and E), the mutant was only associated with half the amount of SUMO1 compared with WT (Fig. 6, B and D). This is significant because the STUbL TOPORS (Liu et al., 2024), which has SUMO1-specific ubiquitination activity (Fig. 9 H), appears only to be recruited to PML bodies once a specific threshold of SUMO1 conjugation has been reached. This is perhaps linked to the need for multiple SUMO-SIM interactions to stably bind TOPORS, which may only occur once a certain "density" of SUMO1 in bodies is reached, effectively creating a switch for TOPORS recruitment. As both TOPORS and RNF4 are required for efficient PML degradation (Fig. 9, D and E), the simplest explanation is that defective SUMO1 modification of L217F PML leads to insufficient TOPORS recruitment, which in turn results in suboptimal polyubiquitin chain synthesis and the accumulation of intermediate conjugates (Fig. 7 D). This hypothesis is supported by the finding that

WT-PML is more stable in cells lacking SUMO1 (Fig. 8, C and D), indicating that SUMO2/3 cannot completely compensate for the loss of SUMO1.

It has also been reported that the STUbL RNF111 (Arkadia) is involved in PML degradation (Erker et al., 2013) and that hybrid SUMO1-SUMO2/3 polymeric chains are better substrates for the ligase than those containing only SUMO2/3 (Sriramachandran et al., 2019). While we found evidence for hybrid SUMO polymers associated with *PML* in vivo, our data show a much less robust recruitment of RNF111 to PML bodies than TOPORS (Fig. 9 A), and siRNA ablation of RNF111 had no effect on WT-PML stability in the U2OS cells studied here (Fig. 9, B–F). It is possible that the concerted action of multiple STUbLs in the degradation of PML may be cell type–dependent and that different cells may rely to different degrees on RNF4, RNF111, and TOPORS activities.

Mutant PML incapable of modification at the major SUMOylation sites K65, K160, and K490 is very stable, although this triple mutant is still detectably SUMO-modified (Lallemand-Breitenbach et al., 2008). Our proteomics data confirmed these to be the major acceptors and found 11 other SUMOylation sites in PML, almost all of which showed increased conjugation with one or both SUMO paralogs in response to arsenic (Fig. S2 E). While apparently "minor," these SUMO sites may not be entirely inconsequential to PML turnover. It is notable that TOPORS has spatially separated SIMs (Fig. 9 G), which, if site-agnostic, may be more influenced by total SUMO1 accumulation over a region, than SUMO1 occupancy at a specific site. Unlike TOPORS, RNF4 shows little preference for a particular SUMO paralog in in vitro assays containing SUMO polymer mimetic substrates (Fig. 9 H). However, structural constraints imposed by the close proximity of its SIM domains (Fig. 9 G) may indirectly impose SUMO paralog preference on RNF4 in vivo. This is because closely arranged SIMs may favor SUMO polymer binding, which form more readily with SUMO2/3 than SUMO1 (Tatham et al., 2001). Therefore, this model suggests two, largely discrete SUMO signals are required before PML can be degraded: SUMO1 in any context and SUMO2/3 in polymers (Fig. 10 B). Future work will aim to test this model, and because the interrelationship between RNF4 and TOPORS appears to have influence beyond PML (Liu et al., 2024), understanding this will likely have broader biological implications.

Inevitably, by revealing more complexity in a system, more questions are raised. How does mutation to single residues in

## A

**% relative to maximum**

| WT | A216T | L217F | | WT+As | A216T+As | L217F+As |
|---|---|---|---|---|---|---|
| 17 | 6 | 10 | SUMO1 conjugated to PML | 100 | 13 | 51 |
| 40 | 24 | 39 | SUMO2 conjugated to PML | 100 | 42 | 75 |
| 37 | 19 | 26 | SUMO conjugated to SUMO | 100 | 31 | 78 |
| 53 | 18 | 31 | Ubiquitin conjugated to PML | 100 | 29 | 99 |
| 15 | 29 | 37 | Ubiquitin conjugated to SUMO | 62 | 33 | 100 |
| 62 | 49 | 59 | Ubiquitin conjugated to ubiquitin | 100 | 49 | 70 |

## B

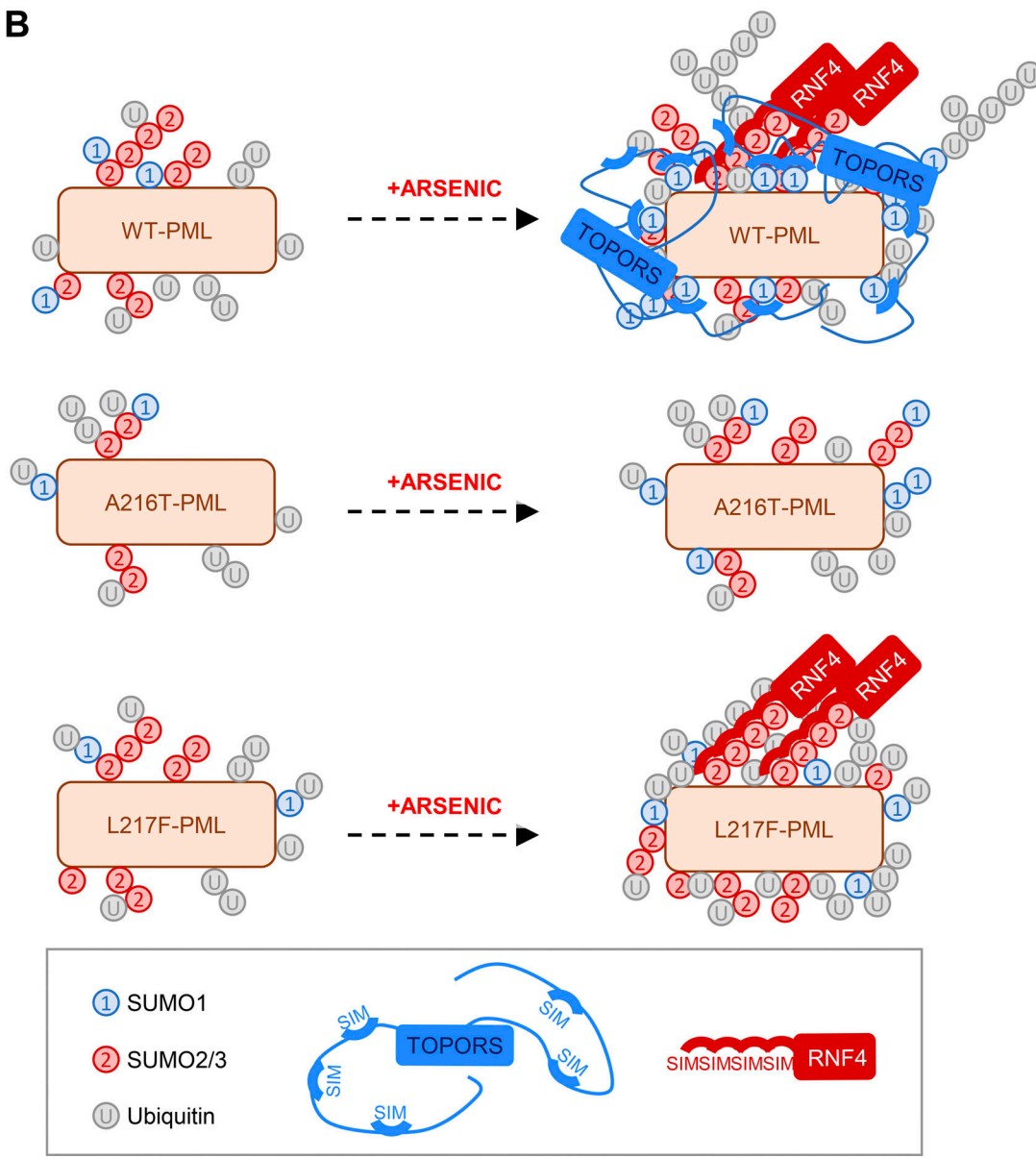

Figure 10. **SUMO paralog-preferential mechanisms for TOPORS and RNF4 cooperate to degrade PML in response to arsenic. (A)** Summary of the proteomics data based on total peptide intensity for each class of peptide indicated. Values are % of the maximum total intensity, and cells are colored based on these. **(B)** The arsenic-induced fates of the two patient-derived PML mutants are the result of suboptimal E3 ligase recruitment. The SIMs in TOPORS are spatially separated in flexible regions of the protein, and at least one has SUMO1 binding preference, while the four SIMs in RNF4 have no SUMO-type preference but are juxtaposed within a region of 35 amino acids. TOPORS will therefore have more conformational flexibility in SUMO binding, than RNF4. SUMO1 lacks an internal conjugation consensus motif so is less likely to form polySUMO conjugates than SUMO2/3. Therefore, TOPORS may preferentially bind multiple, spatially distant SUMO1 molecules, while RNF4 may be more restricted to SUMO2/3 polymers. Arsenic triggers very little change in the SUMOylation status of A216T-PML, which fails to meet the threshold of RNF4 or TOPORS recruitment. A216T does not experience an increase in ubiquitination, so is not degraded by the p97/proteasome system. L2117F PML behaves almost identically to WT-PML for arsenic-induced SUMO2/3 conjugation and SUMO polymer formation, and recruits RNF4. However, an attenuated SUMO1 conjugation response fails to meet the threshold for TOPORS recruitment. Cooperation of TOPORS with RNF4 may be required to synthesize the Ub-Ub chains necessary for recognition by p97, and so, L217F-PML is not efficiently degraded.

PML alter paralog-specific SUMOylation? How do multiple nonredundant STUbLs coordinate in PML ubiquitination? Are different ubiquitin E2 enzymes required for different ubiquitination steps (Liu et al., 2024)? It is expected that further structural and biochemical studies involving clinically relevant PML mutations will help to provide insight into the questions that remain in this pathway that blurs the line between poison and cure.

## Materials and methods
### Preparation of cell extracts and immunoblotting
Adherent cells were washed with phosphate-buffered saline (PBS) and lysed by the addition of 1.2X LDS buffer/60 mM DTT (NuPAGE, Thermo Fisher Scientific) and heated to 70°C for 10 min. Cell lysates were sonicated using a probe sonicator for two bursts of 30 s. Samples were run on NuPAGE 3–8% or 4–12% Bis-Tris precast gels (Thermo Fisher Scientific) and transferred to nitrocellulose membranes. Membranes were blocked in PBS with 5% nonfat milk and 0.1% Tween-20. Primary antibody incubations were performed in PBS with 3% BSA and 0.1% Tween-20. For the HRP-coupled secondary antibody incubations, 5% milk was used instead of the BSA. The signal was detected by Pierce enhanced chemiluminescence (32106; Thermo Fisher Scientific) and X-ray films. Antibodies used in this study were as follows: chicken anti-PML (Hands et al., 2014), mouse anti-alpha tubulin (MA1-80189; Thermo Fisher Scientific), mouse anti-lamin A/C (SAB4200236; Sigma-Aldrich), sheep anti-SUMO-1 (Tatham et al., 2008), rabbit anti-SUMO-2 (4971S; Cell Signaling), rabbit FK2 ubiquitin (BML-PW8810-0500; Enzo), rabbit anti-p97 (ab109240; Abcam), mouse anti-RNF4 (Saito et al., 2014), rabbit anti-TOPORS (ab86383; Abcam). HRP-coupled secondary anti-mouse (A9044), anti-rabbit (A0545), anti-chicken (A9046), and anti-sheep (A3415) antibodies were purchased from Sigma-Aldrich.

### Statistical information
Specific details regarding the statistical analyses can be found in figure legends. Generally, live-cell imaging used multiple fields of view containing multiple cells from the same experiment (see legends). Average PML body size, number of PML bodies per cell, and total intensity of PML bodies per cell were monitored for all cells in all fields. For statistical comparisons among conditions at a fixed time point, data were averaged by field of view. Significance testing used Student's two-tailed unpaired $t$ tests with Welch's correction for unequal variances where necessary.

For colocalization studies using fixed cell immunofluorescence, the percentage of PML bodies colocalized with SUMO1, SUMO2, RNF4, or p97 was manually assessed on a per cell basis from all full cells in multiple fields (see legends for numbers). Nonparametric Kruskal–Wallis tests were performed followed by Dunn's multiple comparisons tests to assess significance of differences comparing WT-PML + arsenic with each mutant + arsenic.

For peptide intensity differences in proteomics data, one-way ANOVA was used for multiple samples testing and two-tailed unpaired $t$ tests were used with Welch's correction where

required, for pairwise comparisons. For protein intensity differences, P values were derived from Brown–Forsythe and Welch ANOVA tests with Dunnett's T3 multiple comparisons test. Peptide and protein intensity values were from four separate YFP-PML purifications.

All parametric tests assumed data to be normally distributed although this was not formally tested.

### Preparation of anti-GFP nanobody magnetic beads
300 mg of Dynabeads M-270 Epoxy (Catalog No. 14302D; Thermo Fisher Scientific) was resuspended in 10 ml of 0.1 M sodium phosphate buffer, pH 7.5, and washed twice using a DynaMag-15 magnet (Catalog No. 12301D; Thermo Fisher Scientific) with 10 ml of the same buffer. 4 mg of the LaG16 nanobody that recognizes GFP (Fridy et al., 2014) in 10 ml of 1 M $(NH_4)_2SO_4$, 0.1 M sodium phosphate buffer, pH 8.0, was added to the beads, which were incubated overnight at 27°C with end-over-end rotation. After coupling was complete, the beads were washed six times with PBS and stored in PBS, 0.1% $NaN_3$ at 4°C.

### Purification of YFP-PML-V from U2OS PML$^{−/−}$ YFP-PML-V cells
For each "replicate" of the proteomics experiment, five 15-cm-diameter plates of ~80% confluent cells were used. Prior to harvesting, growth medium was removed and cells were washed in situ twice with 5 ml PBS/100 mM iodoacetamide. For each dish, cells were scraped into 5 ml PBS + 100 mM iodoacetamide and transferred to a 50-ml tube. A second 2 ml volume of PBS + 100 mM iodoacetamide was used to clean the plate of all cells and pooled with the first 5 ml. This was repeated for all dishes for each replicate and pooled in the same 50-ml tube (~35 ml total). Cells were collected by centrifugation at 400 $g$ for 5 min at 22°C and resuspended in 10 ml 4°C hypotonic buffer including iodoacetamide (10 mM HEPES, pH 7.9, 1.5 mM $MgCl_2$, 10 mM KCl, 0.08% NP-40, 1x EDTA-free protease inhibitor cocktail [Roche], + 100 mM iodoacetamide). These were snap-frozen in liquid nitrogen and stored at –80°C until required. Cells were thawed by tube rotation at 4°C for 30 min. Tubes were then centrifuged at 2,000 $g$, 4°C, for 10 min to sediment nuclei. Supernatants were discarded. Each pellet of nuclei was resuspended in 5 ml ice-cold hypotonic buffer containing 20 mM DTT but without iodoacetamide, transferred to a 15-ml tube, and centrifuged at 2,000 $g$, 4°C for 10 min. The nuclei were washed once more in 5 ml ice-cold hypotonic buffer (without iodoacetamide or DTT), and the supernatant discarded. To digest DNA, nuclei were resuspended in 5 ml "Buffer A" (50 mM Tris, pH 7.5, 150 mM NaCl, 0.2 mM DTT, 1 µg/ml Benzonase, 1x EDTA-free protease inhibitor cocktail [Roche]) and incubated on ice for 15 min. Samples were sonicated in 15-ml tubes on ice (Branson Sonifier) using the small probe, 6 cycles of 30 s at ~50% amplitude with at least 2-min cooling on ice in between cycles. PML bodies should remain intact. Insoluble debris was removed by centrifugation at 1,200 $g$ for 20 min at 4°C, and the supernatant containing PML bodies was transferred to a new 15-ml centrifuge tube. To each 5 ml supernatant, 6 ml "Buffer B" was added (50 mM Tris, pH 7.5, 1% Triton X-100, 0.1% deoxycholic acid, 1x EDTA-free protease inhibitor cocktail [Roche]). To each 11 ml solution, 1 ml volume of a 5% beads:buffer slurry (vol:vol) of

magnetic anti-GFP nanobody beads, pre-equilibrated in 1:1.4 Buffer A:B mix (vol:vol), was added and mixed on a tube roller at 4°C for 16 h (50 µl beads per sample). Using the DynaMag rack (Thermo Fisher Scientific), the beads were extracted from suspension and the supernatant was removed. A sample of this was retained to monitor YFP-PML-V depletion. Beads were then washed with 5 ml "Buffer C" (50 mM Tris, pH 7.5, 500 mM NaCl, 0.2 mM DTT) and the beads extracted. These beads were then resuspended in 1 ml "Buffer D" (50 mM Tris, pH 7.5, 150 mM NaCl, 0.2 mM DTT) and transferred to a new 1.5-ml protein LoBind tube (Eppendorf) and, after bead separation, washed once more with 1 ml Buffer D. One half of this final sample of beads (25 µl beads) was eluted for proteomics analysis by adding 35 µl 1.2X NuPAGE LDS sample buffer/60 mM DTT (Thermo Fisher Scientific), and incubated at 70°C for 15 min with agitation, followed by fractionation by SDS-PAGE. If immunoblot analysis was also required, the second half of the resin (25 µl) would be resuspended in 250 µl 1.2X NuPAGE LDS sample buffer/60 mM DTT (Thermo Fisher Scientific) and incubated at 70°C for 15 min with agitation. Approximately 30 µl would be used per gel lane. If treatment with proteases was required to monitor PML modification status, then the second half volume of beads (25 µl) was divided into four and treated as follows: (1) add 150 µl Buffer D; (2) add 150 µl 250 nM SENP1 in Buffer D; (3) add 150 µl 500 nM USP2 in Buffer D; and (4) add 150 µl 250 nM SENP1/500 nM USP2 in Buffer D. All tubes were agitated for 2 h at 22°C, supernatants removed (and retained for analysis), and beads washed twice with 1 ml Buffer D before elution with 250 µl 1.2X LDS/60 mM DTT (Thermo Fisher Scientific), 70°C, 15 min with agitation.

## Mass spectrometry data acquisition

Elutions from anti-GFP nanobody beads were fractionated on 4–12% polyacrylamide Bis-Tris NuPAGE gels (Thermo Fisher Scientific) using MOPS running buffer. After Coomassie staining and destaining, gels were excised into four slices per lane and tryptic peptides extracted (Shevchenko et al., 2006). Peptides were resuspended in 40 µl 0.1% TFA + 0.5% acetic acid. For the uppermost slice (containing YFP-PML), 20 µl of the tryptic peptide samples was StageTip–purified and eluted in steps of acetonitrile concentration: 20%, 25%, 35%, 50% and 80%, giving 5 trypsin/GluC fractions per sample. Peptides were dried down and each was digested for 16 h at 22°C with 34 µl 7 µg.ml$^{-1}$ GluC in 50 mM ammonium bicarbonate + 5% acetonitrile and then resuspended in 30 µl 0.1% TFA + 0.5% acetic acid. 4 µl of each of the four tryptic digests and 5 trypsin/GluC fractions per lane were analyzed by LC-MS/MS using a Q Exactive mass spectrometer (Thermo Fisher Scientific) coupled to an EASY-nLC 1,000 liquid chromatography system (Thermo Fisher Scientific), using an EASY-Spray ion source (Thermo Fisher Scientific) running a 75 µm × 500 mm EASY-Spray column at 45°C. A 150-min elution gradient with a top 10 data-dependent method was applied for trypsin-only samples, and 90-min gradient with the same settings for trypsin/GluC samples. Full-scan spectra (m/z 300–1,800) were acquired with resolution R = 70,000 at m/z 200 (after accumulation to a target value of 1,000,000 ions with maximum injection time of 20 ms). The 10 most intense ions were fragmented by HCD and measured with a resolution of R = 17,500 at m/z 200 (target value of 500,000 ions and maximum injection time of 60 ms) and intensity threshold of 2.1 × 10$^4$. Peptide match was set to "preferred," a 40-s dynamic exclusion list was applied, and ions were ignored if they had unassigned charge state 1, 8, or >8.

## Mass spectrometry data analysis

For the identification of different modifications, MS data were processed multiple times in MaxQuant (Cox and Mann, 2008; Cox et al., 2011), version 1.6.1.0. All runs included oxidized methionine and acetylated protein N-terminal variable modifications and carbamidomethyl-C as fixed modification. Most parameters were default, although for branched peptide searches, missed cleavages were set to 12, max peptide mass was set to 12,000, and FDR filtering was turned off in favor of manual MS/MS spectrum validation for identifications. For human proteome identifications, FDR was set at 1%. LFQ was switched on, but normalization was skipped. In total, 5 MaxQuant analyses were run (see Data S2 for details). Briefly:

Run 1—SUMO-substrate branched peptides were searched in all samples derived from the uppermost slice using a concatenated protein library method, which used a fasta file with SUMO C-terminal regions fused to the N terminus of putative modified peptides as individual fasta entries (Matic et al., 2008).

Runs 2–4—To search for SUMO adducts in all samples derived from the uppermost slice, multiple variable modifications derived from SUMO1 and SUMO2/3 adducts were considered and searched against a database limited to PML, SUMOs, ubiquitin, and SP100.

Run 5—Identification of copurified proteins and modifications for ubiquitin and phosphorylation were searched using all raw data files (trypsin and trypsin/GluC from all slices) using the human proteome as search space.

Redundancy between concatenated (run1) and variable search (runs2–4) methods for SUMO branched peptides was resolved by carrying forward data from only one identification if the same MS/MS spectra were used for both. For all analyses, peptide/protein intensities in each sample were manually normalized relative to PML peptide intensity in equivalent fractions. In the case of modified peptide-level analysis, raw intensities were used. For protein-level analysis, LFQ intensities were used. To provide a single intensity value per replicate for each modification or protein, normalized protein or peptide intensities were summed for all peptide samples derived from the same lane. Data from multiple evidence of the same SUMOylation at specific sites (by different SUMO or substrate fragments of the same modification) were then calculated by summation of all normalized intensities.

## Cell culture and siRNA-mediated depletion

U2OS cells were grown in high-glucose DMEM (Gibco) supplemented with 10% fetal bovine serum (Labtech), 100 U/ml penicillin, and 100 µg/ml streptomycin (Invitrogen). Cells were transfected with a SMARTpool siRNA containing an equimolar amount of four siRNA duplexes targeting a selected gene (Dharmacon ON-TARGETplus; for Human TOPORS, Human

RNF111, and Human RNF4) to a final concentration of 10 nM, or a non-targeting control duplex (siNT) at the same concentration using Lipofectamine RNAiMAX (Life Technologies) according to the manufacturer's instructions. Cas9 or Cas9-gRNA RNP transfections were performed using a Neon electroporator (briefly, $1 \times 10^6$ cells in 10 µl Buffer R were transfected with 12.5 µM Cas9 or 12.5 µM RNPs using the following settings: 1,230 V, 10 ms, 4 pulses).

## RNA preparation and real-time quantitative PCR
Total RNA was extracted using RNeasy Mini Kit (Qiagen) and treated with the on-column RNase-Free DNase Set (Qiagen) according to the manufacturer's instructions. RNA concentration was then measured using NanoDrop, and 1 µg of total RNA per sample was subsequently used to perform a two-step reverse transcription polymerase chain reaction using random hexamers and First Strand cDNA Synthesis Kit (Thermo Fisher Scientific). Each qPCR contained PerfeCTa SYBR Green FastMix ROX (Quantabio), forward and reverse primer mix (200 nM final concentration), and 6 ng of analyzed cDNA and was set up in triplicates in MicroAmp Fast Optical 96-Well or 384-Well Reaction Plates with Barcodes (Applied Biosystems). The sequences of primers used were as follows: RNF4 (hRNF4_F981 5′-ACTCGTGGAAACTGCTGGAG-3′; hRNF4_F1144 5′-TCATCGTCACTGCTCACCAC-3′), RNF111 (hRNF111_F2719 5′-CAACGCGGGCACATGAAC-3′; hRNF111_R2842 5′-AATTCCCAGTTCCCAGGCAG-3′), TOPORS (hTOPORS_F1374 5′-AGCCACTGTTAGTCAGGCAC-3′; hTOPORS_R1627 5′-CTGGGGTCCTCTCAGCTAGT-3′), TBP (hTBP_F896 5′-TGTGCTCACCCACCAACAAT-3′; hTBP_R1013 5′-TGCTCTGACTTTAGCACCTGTT-3′). Data were collected using QuantStudio 6 Flex Real-Time PCR Instrument and analyzed using corresponding software (Applied Biosystems). Relative amounts of specifically amplified cDNA were calculated using TBP amplicons as normalizers.

## Immunofluorescence analysis
Adherent cells on coverslips were washed three times with PBS prior to fixing with 4% formaldehyde in PBS for 15 mins, then washed again three times with PBS before blocking for 30 min in 5% BSA, 0.1% Tween-20 in PBS. Cells were washed once with 1% BSA, 0.1% Tween-20 in PBS before incubation for 1–2 h with primary antibody in 1% BSA, 0.1% Tween-20 in PBS. These were chicken anti-PML (Hands et al., 2014), sheep anti-p97 (MRC Dundee), sheep anti-SUMO1 (Tatham et al., 2008), and sheep anti-SUMO2 (Tatham et al., 2008) and RNF4 (Saito et al., 2014). After three washes in 1% BSA, 0.1% Tween-20 in PBS cells was incubated with secondary antibody in 1% BSA, 0.1% Tween-20 in PBS for 60 min, washed three times in 1% BSA, 0.1% Tween 20 in PBS, and stained with 0.1 µg/ml DAPI for 2 min. These were Alexa Fluor (Invitrogen) donkey anti-chicken 594 (A78951), donkey anti-sheep 594 (A11016), and donkey anti-chicken 488 (A78948). Coverslips were rinsed twice with PBS, twice with water, and dried before mounting in Mowiol (81381; Sigma-Aldrich). For p97 immunofluorescence analysis, cells were exposed to pre-extraction buffer (25 mM HEPES, pH 7.4, 50 mM NaCl, 3 mM $MgCl_2$, 0.5% Triton X-100, 0.3 M sucrose) for 10 min prior to fixing and processing as described above.

Images were collected using a DeltaVision DV3 widefield microscope using a 60× oil objective (Olympus 1-U2B933; numerical aperture = 1.42) and processed using SoftWoRx (both from Applied Precision). DAPI range signal was detected at Ex 390/Em 435, FITC range signal (YFP) was detected at Ex 480/Em 525, and TRITC range signal was detected at Ex 542/Em 607. Images are presented as maximal intensity projections using Omero software. For quantitative colocalization analysis, the same image settings were applied across all fields. For colocalization of PML with SUMO1, SUMO2, RNF4, or p97, colocalization was defined as any signal above background within the same region as a PML body. Percent colocalization was then calculated for each cell.

## High-content imaging
High-content microscopy was performed using an IN Cell 2,200 microscope (GE Healthcare). IN Cell Analyser Acquisition Software v4.5 was used to acquire three fields of view per well with a 10 or 20× lens (Nikon; numerical aperture for both = 0.45), capturing DAPI, CellMask, and EYFP. Image analysis was performed by IN Cell Developer Toolbox version 1.91 build 2206 (GE Healthcare), using protocols designed to identify EYFP-PML inclusions by multiscale top-hat transformation. All high-content images displayed in this paper were adjusted by applying identical visual parameters for each experiment using ImageJ (NIH). Statistical analysis of the PML data (intensity, count, and total area) was performed using Prism, while for the SUMO1 knockout experiment, the data were only examined for the presence or absence of nuclear bodies to confirm inactivation of both alleles of the SUMO1 genomic locus.

## Live-cell imaging in real time and measurement of fluorescence intensities
For all live-cell imaging experiments, cells were seeded onto µ-slide 8-well glass bottom (80827; ibidi). Immediately prior to imaging, cell medium was replaced with DMEM minus phenol red (31053-038; Gibco) supplemented with 2 mM glutamine and 10% fetal bovine serum. Cells were untreated or treated with 1 µM arsenic. Time-lapse microscopy was performed on a DeltaVision Elite restoration microscope (Cytiva) using a 40× Oil objective (Olympus 1-UB768; numerical aperture = 1.35) and an incubation chamber (Solent Scientific) set to 37°C and 5% $CO_2$ and a cooled charge-coupled device camera (CoolSnap HQ; Roper). SoftWoRx software was used for image collection and dataset deconvolution using the constrained iterative algorithm (Swedlow et al., 1996; Wallace et al., 2001). Using a YFP-specific filter set (Ex 510/10 nm, Em 537/26 nm), 5 z-planes spaced at 2 µm were taken every 15 min. A single bright-field reference image was taken at each time point to monitor cell health. Time courses were presented as maximum intensity projections of deconvolved 3D datasets. Movies were created in OMERO using the deconvolved images.

## Quantitative analysis of PML body composition during live-cell imaging
Batch analysis of time-lapse movies was carried out using ImageJ (Schneider et al., 2012) macros in Fiji (Schindelin et al., 2012).

Briefly: (1) datasets (deconvolved with SoftWoRx) were cropped to remove border artifacts; (2) for each time point, the maximum focus slice was selected according to intensity variance (radius two pixels); (3) autothresholding was performed using a threshold of mean intensity + 3 standard deviations; and (4) ImageJ's built-in "Analyze Particles" function was used to measure PML body count, size, total area, and intensity (Jaffray et al., 2023).

## Generation of SUMO1$^{-/-}$ cells by CRISPR/Cas9

An RNP complex consisting of the guide RNA (sgRNA) 5′-GUG UUCCAAUGAAUUCACUC-3′ specific to SUMO1, tracrRNA (both purchased from IDT), and recombinant Cas9 protein (gift from Dr. Federico Pelisch, University of Dundee, Dundee, UK) was formed according to IDT's protocol and co-electroporated with plasmid vector pEFIRES-P-EYFP-C1 (Ibrahim et al., 2020) into U2OS cells (HTB-96; ATCC) using the Amaxa nucleofection system (Lonza) and the manufacturers' cell type–specific electroporation protocol. 24 h following electroporation, puromycin was added to the cells at 1 µg/ml to enrich the population with cells that received the RNP complex. After 48 h, cells were diluted in DMEM without puromycin and seeded in 96-well plates at a density of 1 cell/well and left for 3–4 wk to form colonies, after which selected clones were reseeded in duplicate in 96-well plates, whereas one replicate was used to identify SUMO1$^{-/-}$ clones by high-content imaging. For this, cells were treated with 1 µM arsenic for 2 h, fixed, immunostained for SUMO1, and imaged by high-content microscopy. SUMO1$^{-/-}$ cells were identified on the basis of the absence of nuclear foci that normally appear as a result of recruiting SUMO1 to PML bodies following exposure to arsenic.

## Constructs for bacterial expression

Sequence coding for human TOPORS (residues 2–574), codon-optimized for expression in *E. coli*, was synthesized by IDT as a gBlocks gene fragment and inserted into pHLTV vector using NEBuilder HiFi DNA Assembly Kit (New England Biolabs). The TOPORS (2-574)_pHLTV construct includes coding sequences for an N-terminal His$_6$-lipoyl domain (residues 2–85 of dihydrolipoyllysine-residue acetyltransferase component of pyruvate dehydrogenase complex from *Geobacillus stearothermophilus*) and a C-terminal Strep-tag II. SIM mutants of TOPORS (2-574) contain the following mutations: SIM1 (V391A, I392A, I394A), SIM2 (C478A, V479A, I480A, V481A), and SIM3 (L494A, V495A, L497A). Full-length *Rattus norvegicus* RNF4 (Tatham et al., 2008) and SIM mutants (Xu et al., 2014) were already available.

4xSUMO1 and 4xSUMO2 constructs encode linear head-to-tail fusions of four copies of SUMO1 (C52A) and SUMO2 (C47A), respectively. The first methionine is mutated to glycine in all four copies of SUMO. 4xSUMO1 and 4xSUMO2 DNA sequences were synthesized by GenScript and subcloned into a pET-30a(+) vector using NdeI and HindIII restriction sites. The expressed protein contains an N-terminal His$_6$-tag, cleavable with TEV protease, and a short C-terminal tag with a single cysteine residue for labeling with a fluorescent dye using maleimide chemistry.

## Protein expression and purification

Lipoyl domain–tagged TOPORS (2-574) was expressed in C41(DE3) *E. coli* cells. Cells were grown in LB medium until OD$_{600}$ reached ~0.5–0.6, followed by rapid cooling of cell cultures in ice-cold water bath. ZnSO$_4$ was added to a final concentration of 10 µM before induction. Protein expression was induced with 100 µM IPTG, followed by incubation of cultures at 20°C with shaking at 200 rpm for ~18 h. Bacterial cells were harvested, resuspended in 50 mM HEPES (pH 8.0), 300 mM NaCl, 0.5 mM TCEP, cOmplete protease inhibitor cocktail (EDTA-free; Roche), and lysed by sonication. The lipoyl domain–tagged TOPORS (2-574) protein was purified by affinity chromatography on Strep-Tactin Sepharose resin (Catalog No. 2-1201-025; IBA Lifesciences) and eluted from the resin with 50 mM HEPES (pH 8.0), 150 mM NaCl, 1 mM TCEP, 2.6 mM desthiobiotin.

Expression and purification of full-length RNF4 were performed as described previously (Plechanovová et al., 2011).

4xSUMO1 and 4xSUMO2 were expressed in BL21(DE3) *E. coli* cells grown in LB medium. Cell cultures were quickly cooled in ice-cold water bath when OD$_{600}$ reached ~0.6. Subsequently, IPTG was added to a final concentration of 100 µM and cell cultures were incubated at 20°C with shaking at 200 rpm for ~18 h. Bacterial cells were harvested, resuspended in 50 mM Tris (pH 7.5), 500 mM NaCl, 10 mM imidazole, 0.5 mM TCEP, cOmplete protease inhibitor cocktail (EDTA-free, Roche), and lysed by sonication. His$_6$-tagged 4xSUMO1 and 4xSUMO2 were purified by Ni-NTA affinity chromatography, followed by cleavage with His$_6$-tagged TEV protease at 4°C overnight. Cleaved His$_6$-tag and TEV protease were removed by second Ni-NTA affinity chromatography. 4xSUMO1 and 4xSUMO2 were further purified by gel filtration chromatography on a HiLoad 16/600 Superdex 200 pg column (Cytiva) equilibrated with 50 mM HEPES (pH 8.0), 150 mM NaCl, 0.5 mM TCEP. Purified 4xSUMO1 and 4xSUMO2 were fluorescently labeled with Alexa Fluor 647 C2-maleimide (Catalog No. A20347; Invitrogen) according to the manufacturer's instructions.

## In vitro ubiquitination assay

In vitro ubiquitination assays were composed of 50 nM His$_6$-Ube1, 0.5 µM UbcH5a, 20 µM ubiquitin, 1.25 µM substrate: 4xSUMO1 or 4xSUMO2 labeled with Alexa Fluor 647 fluorescent dye, and 0.5 µM E3 ligase: lipoyl domain–tagged TOPORS (*Homo sapiens*; residues 2–574) or untagged RNF4 (*R. norvegicus*; full-length). Reactions were 100 µl in volume and were buffered with 50 mM Tris (pH 7.5), 150 mM NaCl, 0.5 mM TCEP, 0.1% (vol/vol) NP-40, 3 mM ATP, and 5 mM MgCl$_2$. Samples were incubated at room temperature, and reactions were started by the addition of ATP. The first time point (0 min) was taken before the addition of ATP. Further time points were taken at 5, 10, and 30 min after the addition of ATP. Reactions (20 µl per time point) were stopped by mixing with an equal volume of reducing SDS-PAGE sample buffer. Samples were analyzed by SDS-PAGE. For in-gel visualization of 4xSUMO substrates, Alexa Fluor 647 fluorescence was detected using Typhoon biomolecular imager (Cytiva). Subsequently, the SDS-PAGE gels were stained with Coomassie blue to visualize all protein bands in the samples.

## Online supplemental material

Fig. S1 shows a proteomics approach to monitor site-specific SUMO and ubiquitin modifications of YFP-PML-V. Fig. S2 shows proteomics data for PML peptides modified by phosphorylation, SUMO1 or SUMO2/3. Fig. S3 shows unmodified counterpart peptide data for PML peptides affected by lysine modifications. Fig. S4 shows site-specific proteomics data of SUMO-SUMO branched peptides. Fig. S5 shows site-specific proteomics data of ubiquitin conjugated to PML. Fig. S6 shows site-specific proteomics data of ubiquitin conjugated to SUMOs. Fig. S7 shows TOPORS knockout inhibits arsenic-induced PML degradation, and SIM2 is the major SUMO1-interacting element. Video 1 shows time-lapse YFP-PML fluorescence microscopy of untreated U2OS PML$^{-/-}$ cells +YFP-WT-PML-V. Video 2 shows time-lapse YFP-PML fluorescence microscopy of 1 μM arsenic-treated U2OS PML$^{-/-}$ cells +YFP-WT-PML-V. Video 3 shows time-lapse YFP-PML fluorescence microscopy of untreated U2OS PML$^{-/-}$ cells +YFP-A216T-PML-V. Video 4 shows time-lapse YFP-PML fluorescence microscopy of 1 μM arsenic-treated U2OS PML$^{-/-}$ cells +YFP-A216T-PML-V. Video 5 shows time-lapse YFP-PML fluorescence microscopy of untreated U2OS PML$^{-/-}$ cells +YFP-L217F-PML-V. Video 6 shows time-lapse YFP-PML fluorescence microscopy of 1 μM arsenic-treated U2OS PML$^{-/-}$ cells +YFP-L217F-PML-V. Data S1 shows a summary of the proteomics data for posttranslational modifications and proteins associated with purified YFP-PML bodies as described in Fig. S1. Data S2 shows details of the multiple MaxQuant runs made to generate the data in Data S1.

## Data availability

The mass spectrometry proteomics data underlying Fig. 6, Fig. 7, Fig. 9, Fig. S1, Fig. S2, Fig. S3, Fig. S4, Fig. S5, and Fig. S6 have been deposited to the ProteomeXchange Consortium via the PRIDE (Perez-Riverol et al., 2022) partner repository with the dataset identifier PXD051743.

## Acknowledgments

The authors would like to thank Professor Takeshi Urano (Shimane University School of Medicine) for the kind gift of the RNF4 monoclonal antibody. pHLTV vector was a kind gift from Dr. Mark Allen (University of Oxford, Oxford, UK).

Work in the Hay laboratory was supported by an Investigator Award from Wellcome Trust (217196/Z/19/Z) and a Programme grant from Cancer Research UK (DRCRPG-May23/100003). Work in the Mailand laboratory was supported by a grant NNF18OC0030752 (Novo Nordisk Foundation). Open Access funding provided by the University of Dundee.

Author contributions: E.G. Jaffray: data curation, formal analysis, investigation, methodology, validation, visualization, and writing—original draft. M.H. Tatham: data curation, formal analysis, investigation, methodology, validation, visualization, and writing—original draft, review, and editing. B. Mojsa: formal analysis, investigation, resources, visualization, and writing—review and editing. A. Plechanovova: investigation, methodology, visualization, and writing—review and editing. A. Rojas-Fernandez: conceptualization, investigation, methodology, and writing—review and editing. J.C.Y. Liu: resources and writing—review and editing. N. Mailand: resources, supervision, and writing—review and editing. A.F.M. Ibrahim: formal analysis, investigation, methodology, and visualization. G. Ball: formal analysis, software, and writing—review and editing. I.M. Porter: formal analysis, investigation, software, visualization, and writing—review and editing. R.T. Hay: conceptualization, funding acquisition, investigation, methodology, project administration, resources, supervision, validation, visualization, and writing—original draft, review, and editing.

Disclosures: The authors declare no competing interests exist.

Submitted: 20 July 2024

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

# Supplemental material

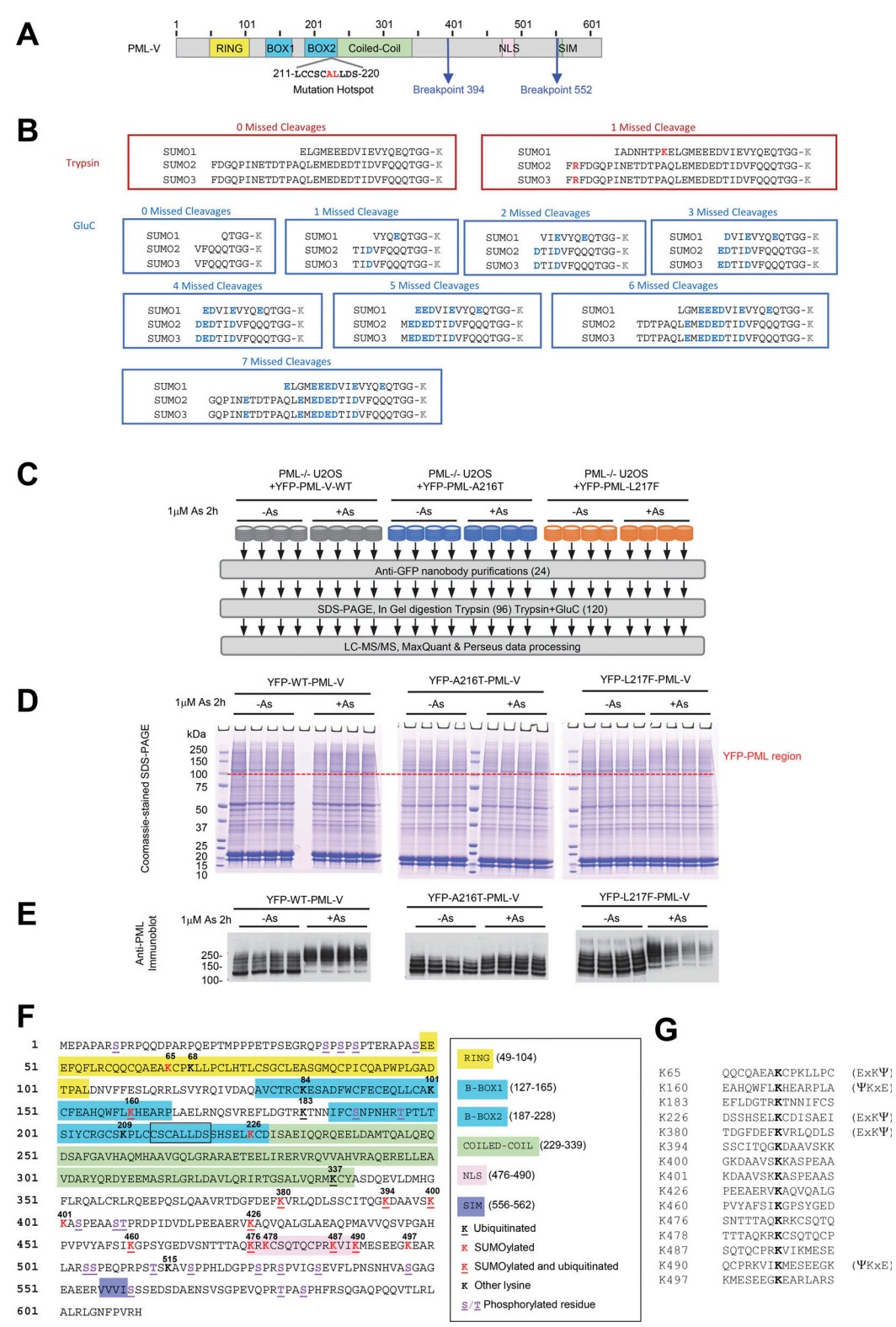

Figure S1. **Proteomics approach to monitor site-specific SUMO and ubiquitin modifications of YFP-PML-V.** Related to Fig. 1, Fig. 6, and Fig. 7. **(A)** Schematic depiction of PML-V primary structure indicating the major domains, regions of common mutations associated with arsenic resistance, and the common breakpoints leading to PML-RARA fusions. **(B)** Remnants of SUMO C termini attached to substrate lysine residues (K) when conjugates are cleaved by trypsin, or GluC. **(C)** Overview of the proteomics experiment to monitor PML modifications at the site level. **(D)** Coomassie-stained gel showing anti-GFP nanobody purifications as indicated from C. The upper region containing YFP-PML-V and its modified forms is indicated. **(E)** Anti-PML immunoblot of a fraction of the anti-GFP nanobody purifications for each replicate. **(F)** Positions of SUMO and ubiquitin conjugation sites in PML-V identified in this study. The mutational hotspot found in arsenic-insensitive forms is boxed. **(G)** 15 residue sequence windows around sites of SUMO and/or ubiquitin attachment identified in PML-V.++. Source data are available for this figure: SourceData FS1.

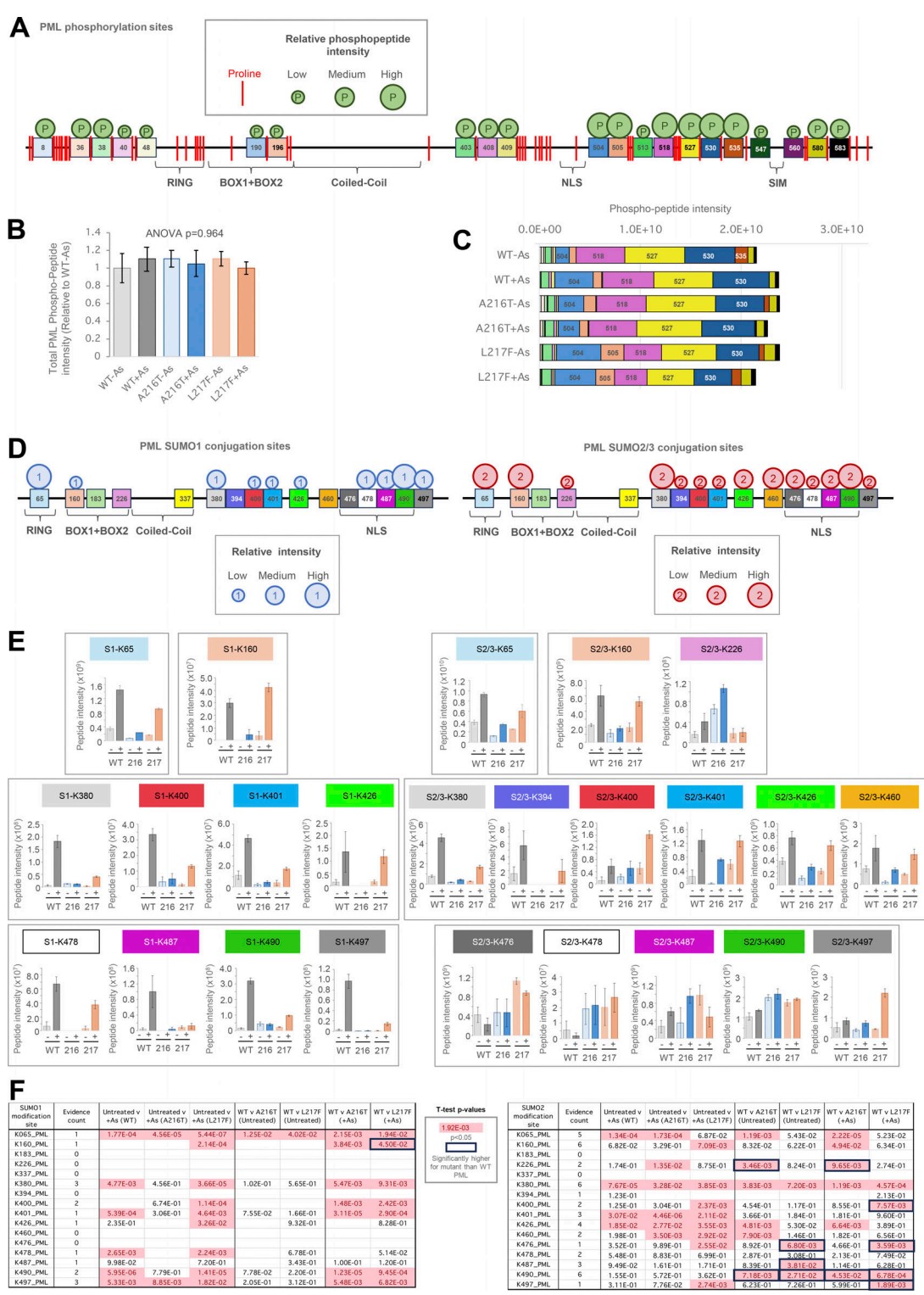

Figure S2. **Proteomics data for PML peptides modified by phosphorylation, SUMO1 or SUMO2/3.** Relating to Fig. S1 and Fig. 6. **(A)** Schematic representation of the PML-V sequence with sites of phosphorylation identified in this study indicated. Relative phosphopeptide intensity at each site is approximated by circle size. Positions of proline residues are shown by red bars. **(B)** Average (column) and SEM (bars) for the sum all phosphopeptide intensities identified in each sample prepared in the proteomics experiment described in Fig. S1. Ordinary one-way ANOVA P value is indicated. Number of purifications, *n* = 4. **(C)** As in B but showing the contribution of individual sites to overall peptide intensities. **(D)** Schematic representation of the PML-V sequence indicating sites of SUMO1 (left) and SUMO2/3 (right) modification identified in this study. Relative peptide intensity at each site is approximated by circle size. **(E)** Column charts of average intensity values for the SUMO1 (left) and SUMO2/3 (right) branched peptides diagnostic of PML modification at the indicated sites. Error bars show SEM (*n* = 4). **(F)** *t* test summary for the pairwise comparisons indicated for each SUMO-PML modification. Welch's correction was applied for heteroskedastic data. Red entries are P < 0.05, and boxed values show sites where SUMO-PML peptide intensities were significantly higher for the mutant than the WT (last four columns). The number of different peptides used for each site is indicated (evidence count).

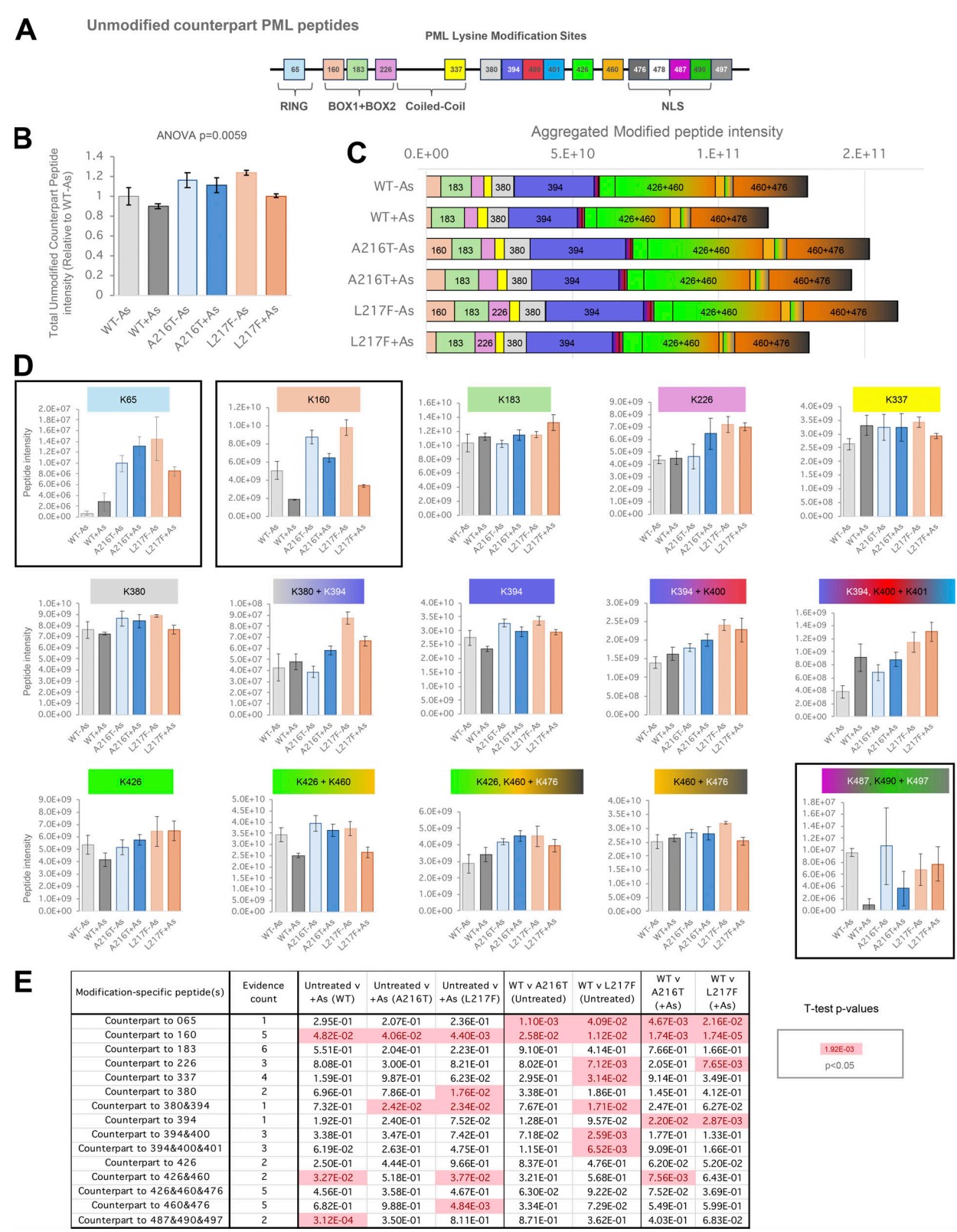

**Figure S3.** **Unmodified counterpart peptide data for PML peptides affected by lysine modifications.** Related to Fig. 6 and Fig. 7. **(A)** PML schematic showing positions of lysines modified by SUMO or ubiquitin. **(B)** Average (column) and SEM (bars) for the sum all unmodified counterpart peptide intensities identified in each YFP-PML purification. Ordinary one-way ANOVA P value is indicated. Number of purifications, *n* = 4. **(C)** Total unmodified counterpart peptide intensity data broken down by site. **(D)** Average intensity values for the sum of all counterpart peptides relating to modification at the indicated lysine. Boxed plots relate to the canonical major SUMO acceptor lysines. Error bars show SEM (number of purifications, *n* = 4). **(E)** *t* test summary for the pairwise comparisons indicated for each SUMO-PML modification. Welch's correction was applied for heteroskedastic data. Red entries are P < 0.05. The number of different counterpart peptides used for each site is indicated as evidence count.

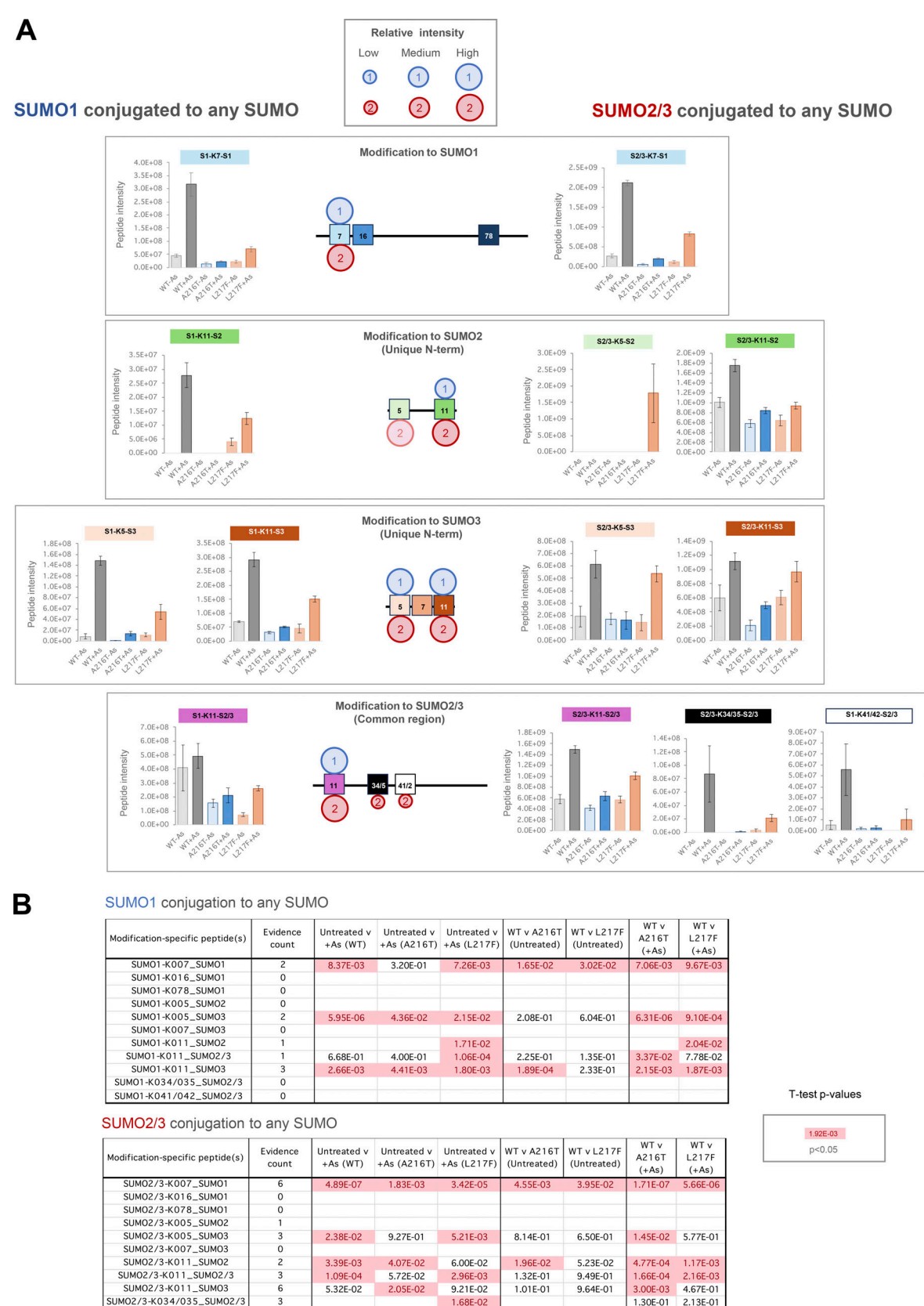

Figure S4.  **Site-specific proteomics data of SUMO-SUMO branched peptides.** Related to Fig. 6, F–H. **(A)** Column charts of average intensity values for the branched peptides diagnostic of SUMO-SUMO linkages via the indicated sites. Error bars show SEM. Number of purifications, *n* = 4. **(B)** *t* test summary for the two-tailed pairwise comparisons indicated for each SUMO1 (upper) or SUMO2/3 (lower) modifications to any SUMO. Welch's correction was applied for heteroskedastic data. Red entries are P < 0.05. The number of different peptides used for each site is indicated (evidence count).

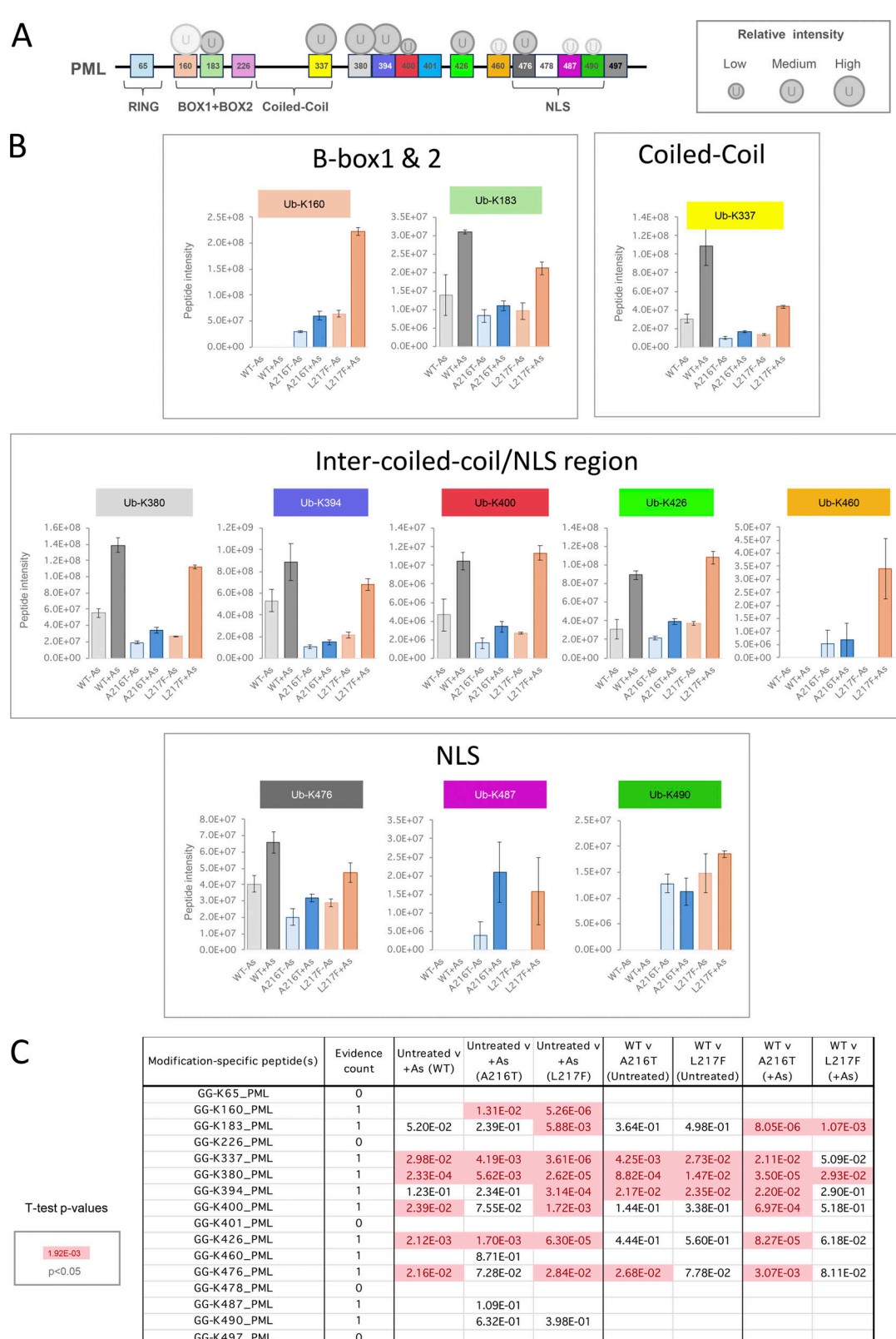

Figure S5.   **Site-specific proteomics data of ubiquitin conjugated to PML.** Related to Fig. 7, A and B. **(A)** PML schematic showing positions of lysines modified by ubiquitin. Circle size approximates peptide intensity, and pale circles were not detected in WT-PML purifications. **(B)** Column charts of average intensity values for the branched peptides diagnostic of PML ubiquitination at the indicated sites. Error bars show the SEM. Number of purifications, $n = 4$. **(C)** $t$ test summary for the two-tailed pairwise comparisons indicated for each ubiquitin-PML modification. Welch's correction was applied for heteroskedastic data. Red entries are $P < 0.05$. Schematic shows sites in relation to PML sequence and functional domains. The number of different peptides used for each site is indicated (evidence count).

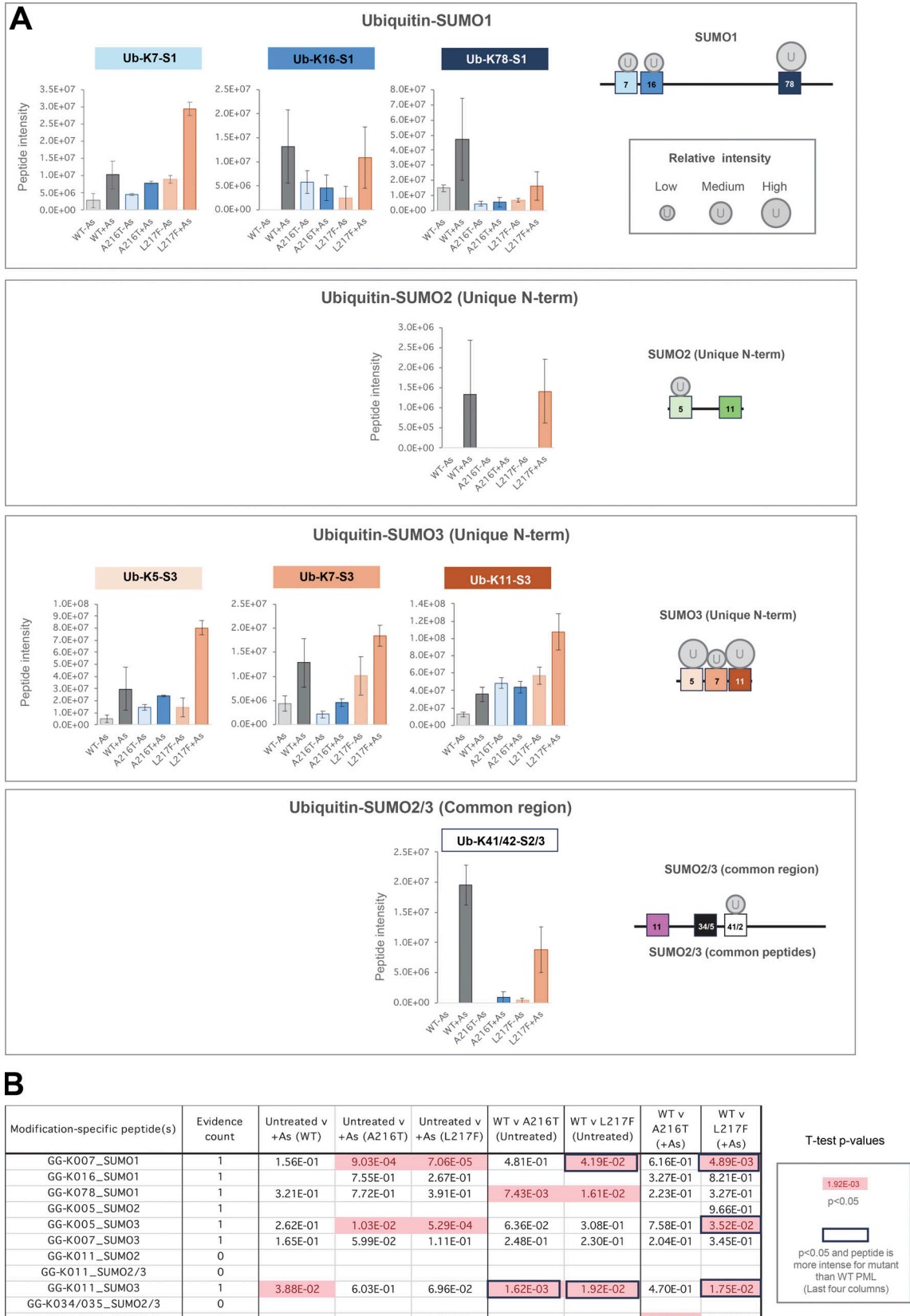

Figure S6.   **Site-specific proteomics data of ubiquitin conjugated to SUMOs.** Relating to Fig. 7, C and D. **(A)** Column charts of average intensity values for the branched peptides diagnostic of ubiquitin-SUMO linkages via the indicated sites. Error bars show the SEM. Number of purifications, *n* = 4. **(B)** *t* test summary for the two-tailed pairwise comparisons indicated for each ubiquitin-SUMO modification. Welch's correction was applied for heteroskedastic data. Red entries are P < 0.05. Boxed values indicate those sites where the peptide intensities were significantly higher for the mutant PML than WT.

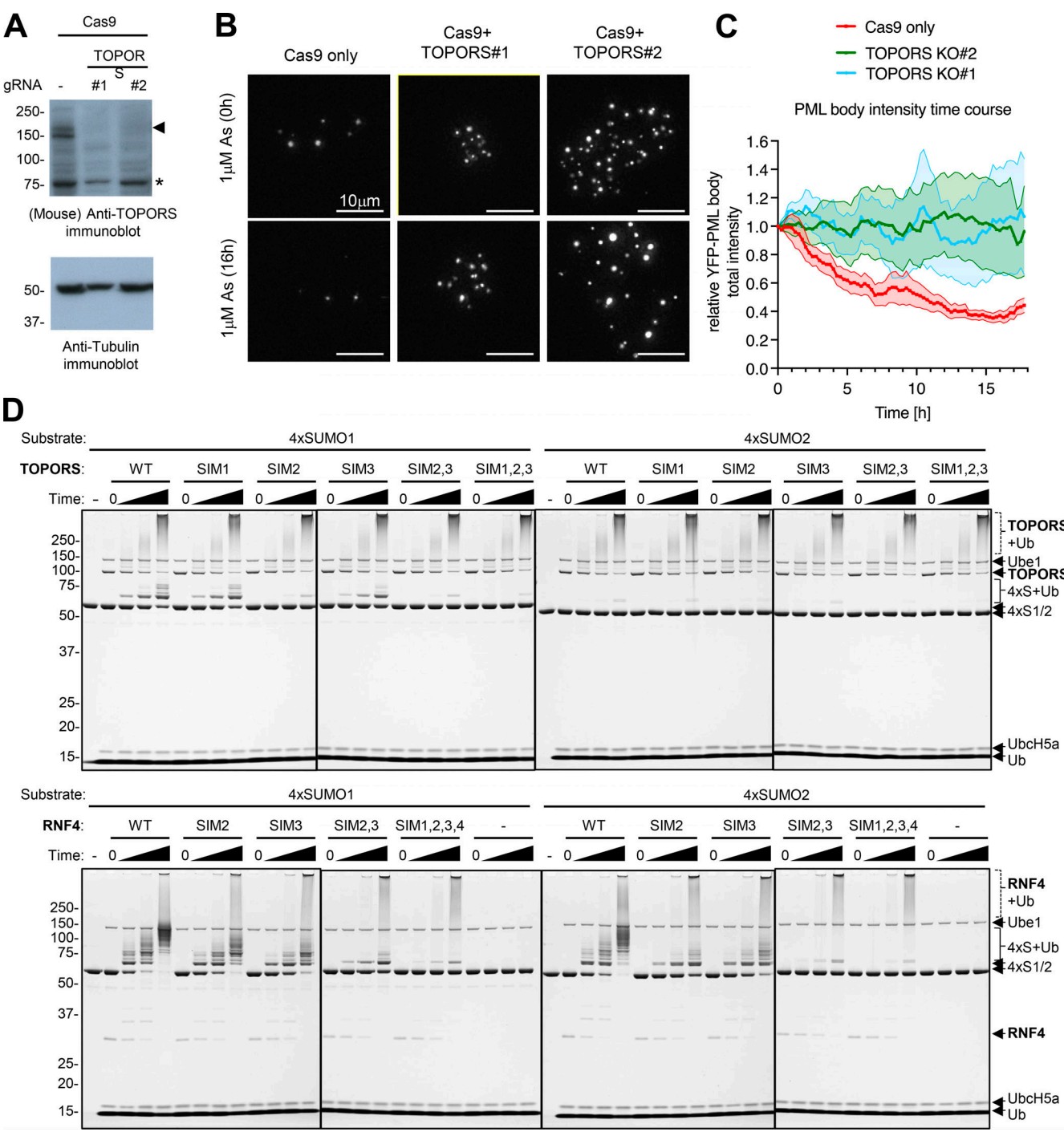

Figure S7. **TOPORS knockout inhibits arsenic-induced PML degradation, and SIM2 is the major SUMO1-interacting element.** Related to Fig. 9. **(A)** Immunoblots for TOPORS and tubulin from crude cell extracts of cells transfected with Cas9 and TOPORS #1 and TOPORS #2 guide RNA containing RNPs. **(B)** Representative microscopy images of U2OS YFP-PML-V WT cells transfected with Cas9 only or Cas9 with two different guides for TOPORS at t = 0 h and t = 16 h after 1 µM arsenic treatment. **(C)** Average PML body intensity over an 18-h time course of arsenic exposure in cells transfected with Cas9 only or Cas9-TOPORS RNPs. Solid lines represent average intensity normalized to t = 0 h, and shaded areas are SEMs (*n* = 10 fields of view). **(D)** Coomassie-stained whole gel images for the 4xSUMO1 and 4xSUMO2 ubiquitination assays using TOPORS and RNF4 SIM mutants as shown in Fig. 9 H. Source data are available for this figure: SourceData FS7.

Video 1.   **Time-lapse YFP-PML fluorescence microscopy of untreated U2OS PML$^{-/-}$ cells +YFP-WT-PML-V.** Images at 15-min intervals over 15 h. The scale bar is 5 µm.

Video 2.   **Time-lapse YFP-PML fluorescence microscopy of 1 µM arsenic-treated U2OS PML$^{-/-}$ cells +YFP-WT-PML-V.** Images at 15-min intervals over 15 h. The scale bar is 5 µm. 0- and 15-h stills are shown in Fig. 1 C.

Video 3.   **Time-lapse YFP-PML fluorescence microscopy of untreated U2OS PML$^{-/-}$ cells +YFP-A216T-PML-V.** Images at 15-min intervals over 15 h. The scale bar is 5 µm.

Video 4.   **Time-lapse YFP-PML fluorescence microscopy of 1 µM arsenic-treated U2OS PML$^{-/-}$ cells +YFP-A216T-PML-V.** Images at 15-min intervals over 15 h. The scale bar is 5 µm. 0- and 15-h stills are shown in Fig. 1 C.

Video 5.   **Time-lapse YFP-PML fluorescence microscopy of untreated U2OS PML$^{-/-}$ cells +YFP-L217F-PML-V.** Images at 15-min intervals over 15 h. The scale bar is 5 µm.

Video 6.   **Time-lapse YFP-PML fluorescence microscopy of 1 µM arsenic-treated U2OS PML$^{-/-}$ cells +YFP-L217F-PML-V.** Images at 15-min intervals over 15 h. The scale bar is 5 µm. 0- and 15-h stills are shown in Fig. 1 C.

**Provided online are Data S1 and Data S2. Data S1 shows summary of the proteomics data for posttranslational modifications and proteins associated with purified YFP-PML bodies as described in Fig. S1. Data S2 shows details of the multiple MaxQuant runs made to generate the data in Data S1.**

