## [Peer Review File · The Journal of Cell Biology]

PML Mutants from Arsenic-Resistant Patients Reveal SUMO1-TOPORS and SUMO2-RNF4 Degradation Pathways

Ellis Jaffray, Michael Tatham, Barbara Mojsa, Anna Plechanovova, Alejandro Rojas-Fernandez, Julio Liu, Niels Mailand, Adel Ibrahim, Graeme Ball, Iain Porter, and Ronald Hay

Corresponding Author(s): Ronald Hay, University of Dundee

Review Timeline:

Submission Date:	2024-07-20
Editorial Decision:	2024-09-05
Revision Received:	2025-01-31
Editorial Decision:	2025-02-17
Revision Received:	2025-03-01

Monitoring Editor: Ira Mellman

Scientific Editor: Dan Simon

Transaction Report:

DOI: <https://doi.org/10.1083/jcb.202407133>

September 5, 2024

Re: JCB manuscript #202407133

Dr. Ronald T Hay
University of Dundee
Centre for Gene Regulation and Expression Sir James Black Centre Dow Street
Dundee, Scotland DD15EH
United Kingdom

Dear Dr. Hay,

Thank you for submitting your manuscript entitled "PML Mutants from Arsenic-Resistant Patients Reveal SUMO1-TOPORS and SUMO2-RNF4 Degradation Pathways." The manuscript was assessed by expert reviewers, whose comments are appended to this letter. We invite you to submit a revision if you can address the reviewers' key concerns, as outlined here.

You will see that the reviewers are enthusiastic about your study. They ask for a couple of additional experiments, colocalization quantifications, and controls to strengthen the conclusions and clarify some points. These all seem straightforward to us and we hope you will be able to address all of the comments.

GENERAL GUIDELINES:

Text limits: Character count for an Article is < 40,000, not including spaces. Count includes title page, abstract, introduction, results, discussion, and acknowledgments. Count does not include materials and methods, figure legends, references, tables, or supplemental legends.

Figures: Articles may have up to 10 main text figures. Figures must be prepared according to the policies outlined in our Instructions to Authors, under Data Presentation, <https://jcb.rupress.org/site/misc/ifora.xhtml>. All figures in accepted manuscripts will be screened prior to publication.

Supplemental information: There are strict limits on the allowable amount of supplemental data. Articles may have up to 5 supplemental figures. Up to 10 supplemental videos or flash animations are allowed. A summary of all supplemental material should appear at the end of the Materials and methods section.

Please note that JCB now requires authors to submit Source Data used to generate figures containing gels and Western blots with all revised manuscripts. This Source Data consists of fully uncropped and unprocessed images for each gel/blot displayed in the main and supplemental figures. Since your paper includes cropped gel and/or blot images, please be sure to provide one Source Data file for each figure that contains gels and/or blots along with your revised manuscript files. File names for Source Data figures should be alphanumeric without any spaces or special characters (i.e., SourceDataF#, where F# refers to the associated main figure number or SourceDataFS# for those associated with Supplementary figures). The lanes of the gels/blots should be labeled as they are in the associated figure, the place where cropping was applied should be marked (with a box), and molecular weight/size standards should be labeled wherever possible. Source Data files will be made available to reviewers during evaluation of revised manuscripts and, if your paper is eventually published in JCB, the files will be directly linked to specific figures in the published article.

The typical timeframe for revisions is three to four months. If you anticipate any difficulties in meeting this aforementioned revision time limit, please contact us and we can work with you to find an appropriate time frame for resubmission. Please note that papers are generally considered through only one revision cycle, so any revised manuscript will likely be either accepted or rejected.

Thank you for this interesting contribution to Journal of Cell Biology. You can contact us at the journal office with any questions at cellbio@rockefeller.edu.

Sincerely,

Ira Mellman, PhD
Monitoring Editor
Journal of Cell Biology

Dan Simon, PhD
Scientific Editor
Journal of Cell Biology

Reviewer #1 (Comments to the Authors (Required)):

This is a very interesting and thorough study, building on the long-lasting interest of the Hay lab to establish that PML mutants from arsenic-resistant patients are due to defects in SUMO1-TOPORS and SUMO2/3-RNF4 degradation pathways. In general, this is an excellent study. A few points remain to be addressed as outlined below.

1. Please verify whether any mutations in TOPORS are associated with APL and therapy resistance.
2. Figure 7E. Hybrid chains of SUMO1/SUMO2/3 are missing. How well are they modified by TOPORS and RNF4?
3. Figure 7E. Please include proper SIM mutants of TOPORS and RNF4.
4. Figure 7 and Figure S13 - Please verify the knockdowns of RNF4 and TOPORS and knockout of TOPORS by Western.
5. Microscopy. Proper statistics of microscopy experiments is provided in Figure 1 and Figure 3, but is missing in Figure 2B and Figure 7D. Please add statistics for Figure 2B and Figure 7D as well.
6. Potential role of STUbL RNF111. Figure S13C shows a single descriptive panel on RNF111. RNF111 is now omitted from the model, whereas Erker et al. 2013 found an important role for this STUbL, so I believe that based on the current data, RNF111 can't be omitted from the model. For a comprehensive picture, it would be important to establish whether RNF111 is required for arsenic-induced degradation of PML in cells or not. Testing RNF111 knockout cells would be conclusive for this purpose.

Reviewer #2 (Comments to the Authors (Required)):

This manuscript investigates the molecular mechanisms underlying arsenic resistance in acute promyelocytic leukemia (APL), focusing on the degradation pathways of PML mutants derived from arsenic-resistant patients. The study reveals distinct roles for SUMO1 and SUMO2/3 in the degradation of PML, mediated by the SUMO-targeted ubiquitin ligases TOPORS and RNF4. Through a combination of biochemical, proteomic, and cell-based assays, the authors explore how these PML mutants resist arsenic-induced degradation, which is critical for understanding treatment resistance in APL.

The manuscript provides valuable insights into the role of SUMO paralogs in the degradation of PML, showing that each SUMO paralog preferentially engages different ubiquitin ligases (TOPORS and RNF4), which are essential for the degradation process. The authors integrated a wide range of experimental techniques, including high-content microscopy, proteomics, immunoblotting, and CRISPR-Cas9 gene editing to characterize two common PML mutants (A216T and L217F) resistant to arsenic and further understand how these modifications affect their degradation. The findings have clear clinical implications, particularly for patients with APL who relapse due to arsenic resistance. Understanding the molecular basis of resistance can inform the development of more effective therapeutic strategies.

While the study is strong in its novelty, thorough characterization, and clinical relevance, it would benefit from additional experiments to support claims. Below are specific comments to strengthen the overall impact of the research.

1. YFP-L217FPML-V recruitment of RNF4 (Figure 3B). The enhanced recruitment of RNF4 upon As treatment is unclear. Could the authors provide a more distinct image or include statistical measurements to better support this observation?
2. Engagement of p97 by YFP-L217FPML-V (Line 306). The authors state that YFP-L217FPML-V fails to engage p97 despite being SUMOylated and recruiting RNF4 (6h 1uM As). In Figure 3A, the lower signals for ubiquitin and RNF4 compared to WT suggest that a certain level of ubiquitination is necessary for p97 engagement. Could the authors clarify whether this observation is influenced by the incubation time or the concentration of As?
3. Ubiquitin Accumulation and p97 Recognition (Line 463). The authors suggest that As induces ubiquitin accumulation on WT-PML, facilitating its degradation, but not on the mutants. However, the bar plots in Figure 5G show modest changes in Ub-Ub conjugates (< 2-fold), and the data do not provide insights into the polyubiquitin chain types (K11 or K48) recognized by proteasome or p97 receptors. Could the authors elaborate on the conclusions that can be drawn regarding the signal recognized by p97?
4. Pearson Correlation Analysis for Figure 3B. Conducting a Pearson correlation coefficient analysis between SUMO1 and PML, and between SUMO2 and PML in the immunofluorescence experiments would enhance the clarity and quantitative analyses of the data.
5. In Vitro Ubiquitination Assays (Line 524, Figure 7E). The in vitro ubiquitination assays indicate that 4xSUMO1 is preferentially ubiquitinated in the presence of TOPORS, while both 4xSUMO1 and 4xSUMO2 are modified by RNF4. These observations, though intriguing, raise mechanistic questions. For example, what is the precise nature of the SUMO1 or mixed SUMO2/3-SUMO1 chains recognized by TOPORS? The use of 4xSUMO1 in vitro has not been confirmed on PML, leaving this point unresolved.
6. TOPORS Colocalization Analysis for Figure 7. Including an immunofluorescence analysis of TOPORS colocalization across all three PML variants, similar to the RNF4 colocalization shown in Figure 3B, would be beneficial. Demonstrating TOPORS and PML colocalization in WT and SUMO1 ^{-/-} cells would further validate the selective recruitment of TOPORS.
7. Clarifying TOPORS and RNF4 Cooperation. The study does not fully elucidate how TOPORS and RNF4 cooperate or compensate for each other in PML degradation. Additional experiments detailing their individual contributions and potential compensatory mechanisms would strengthen the section 'L217F PML fails to recruit TOPORS in response to arsenic'.

Minor points:

1. Line 242, These SUMO-ubiquitin species released by SENP1 were also detected using a ubiquitin antibody (Fig. S3B).
2. Line 350 and Figure 4B, could the authors clarify whether the changes in relative intensity correspond to aggregate numbers for all ubiquitin or SUMO1/2/3 sites, considering that modifications can take place at multiple residues?
3. Loading Controls: Several Western blots lack loading controls. Including these would improve the reliability of the data presented.

Editor requests

Text limits: Character count for an Article is < 40,000, not including spaces. Count includes title page, abstract, introduction, results, discussion, and acknowledgments. Count does not include materials and methods, figure legends, references, tables, or supplemental legends.

Character count in the revised script for these sections is 39720

Figures: Articles may have up to 10 main text figures. Figures must be prepared according to the policies outlined in our Instructions to Authors, under Data Presentation, <https://jcb.rupress.org/site/misc/ifora.xhtml>. All figures in accepted manuscripts will be screened prior to publication.

We have reformatted the manuscript to include the new data and to try to be consistent with these policies. We now have 10 main figures (increased from 8) and 7 supplementary figures (reduced from 13).

Supplemental information: There are strict limits on the allowable amount of supplemental data. Articles may have up to 5 supplemental figures. Up to 10 supplemental videos or flash animations are allowed. A summary of all supplemental material should appear at the end of the Materials and methods section.

We have added a summary of supplementary data as requested but again note that we have reduced the supplementary figures down from 13 to 7.

We have included pdf files containing uncropped images for all immunoblot and immunofluorescence images that were cropped in the figures.

Reviewer #1 (Comments to the Authors (Required)):

1. Please verify whether any mutations in TOPORS are associated with APL and therapy resistance.

Search of the Catalogue of Somatic Mutations in Cancer database (cancer.sanger.ac.uk/cosmic) did not reveal any consistent links between TOPORS and APL. Searches on omim.org and hgmd.cf.ac.uk only report Retinitis Pigmentosa as diseases associated with TOPORS mutations. To our knowledge no mutations to STUbLs associated with PML turnover have been implicated in arsenic resistance in APL patients, although a large, genome-wide study of APL patients has not been conducted.

2. Figure 7E. Hybrid chains of SUMO1/SUMO2/3 are missing. How well are they modified by TOPORS and RNF4?

Due to the strong correlation between TOPORS and SUMO1 abundance in PML bodies (new Fig. 9A) we decided to consider only SUMO1 or SUMO2/3 in isolation in our biochemical assays. While we agree with the reviewer that the potential role of hybrid chains in TOPORS activity is interesting, a thorough investigation would need

to consider many variables including site of linkage, polymer length and SUMO paralog type. Although we can determine which linkages are present, technology for determining precise chain topology for SUMO (and ubiquitin) is presently not available. Thus, any hybrids generated would not necessarily be representative of the situation in vivo. We suggest that a study of hybrid chains is beyond the scope of the present publication. We hope the new work assessing TOPORS SIM mutants (see point 3), satisfies the reviewer's desire for more biochemical details of TOPORS SUMO interactions.

3. Figure 7E. Please include proper SIM mutants of TOPORS and RNF4.

We have repeated our biochemical assays for 4xSUMO1 and 4xSUMO2 modification but this time including RNF4 and TOPORS SIM mutants (new Fig. 9H and Fig. S7D). This has shown that mutation to SIM2 has the greatest single effect on TOPORS activity in vitro. This adds important findings that we think have strengthened the paper.

4. Figure 7 and Figure S13 - Please verify the knockdowns of RNF4 and TOPORS and knockout of TOPORS by Western.

To accommodate the reviewer's request for RNF111 analysis (point 6 below) we have repeated the time-resolved study of the effects of STUbL knock-down on PML degradation, this time including siRNA ablation of RNF111. These new data, including evidence of knock-down is shown in new Fig. 9B-F.

5. Microscopy. Proper statistics of microscopy experiments is provided in Figure 1 and Figure 3, but is missing in Figure 2B and Figure 7D. Please add statistics for Figure 2B and Figure 7D as well.

We have added statistical analysis to figures 3, 4 and 9.

6. Potential role of STUbL RNF111. Figure S13C shows a single descriptive panel on RNF111. RNF111 is now omitted from the model, whereas Erker et al. 2013 found an important role for this STUbL, so I believe that based on the current data, RNF111 can't be omitted from the model. For a comprehensive picture, it would be important to establish whether RNF111 is required for arsenic-induced degradation of PML in cells or not. Testing RNF111 knockout cells would be conclusive for this purpose.

We have undertaken studies using siRNA knockdown of RNF111, RNF4 and TOPORS to monitor affects to PML body degradation (new Fig. 9 B-F). This showed that in the U2OS cells used in this study, RNF111 knock-down did not affect arsenic-induced PML turnover, while RNF4 and TOPORS did. We think that the inconsistency with the existing literature may be cell-line related, and have included discussion in the text to this effect.

Reviewer #2 (Comments to the Authors (Required)):

1. YFP-L217FPML-V recruitment of RNF4 (Figure 3B). The enhanced recruitment of RNF4 upon As treatment is unclear. Could the authors provide a more distinct image or include statistical measurements to better support this observation?

We have provided monochrome images for these panels to enhance clarity, and where necessary, new fields. We have also included statistical analysis of the data in new Fig. 4.

2. Engagement of p97 by YFP-L217FPML-V (Line 306). The authors state that YFP-L217FPML-V fails to engage p97 despite being SUMOylated and recruiting RNF4 (6h 1 μ M As). In Figure 3A, the lower signals for ubiquitin and RNF4 compared to WT suggest that a certain level of ubiquitination is necessary for p97 engagement. Could the authors clarify whether this observation is influenced by the incubation time or the concentration of As?

We agree that there is likely to be a threshold of ubiquitination that must be crossed before PML is extracted from nuclear bodies by p97 and that dose/duration of arsenic may influence this. However, concentrations of arsenic greater than 1 μ M for 24h have been shown to negatively affect cell viability (Kang et al 2003 *Experimental and Molecular Medicine*, Vol.35, No.2, 83-90). Therefore, while the effect of higher or longer arsenic doses on PML mutant degradation is an interesting question, we prefer to limit exposure to 1 μ M for 24h to avoid stress responses complicating data interpretation. We have added text to the results section to explain our choice of arsenic dose.

3. Ubiquitin Accumulation and p97 Recognition (Line 463). The authors suggest that As induces ubiquitin accumulation on WT-PML, facilitating its degradation, but not on the mutants. However, the bar plots in Figure 5G show modest changes in Ub-Ub conjugates (< 2-fold), and the data do not provide insights into the polyubiquitin chain types (K11 or K48) recognized by proteasome or p97 receptors. Could the authors elaborate on the conclusions that can be drawn regarding the signal recognized by p97?

We agree the data showing ubiquitin-ubiquitin linkage accumulation on PML does not show a striking increase with arsenic treatment. Our previous work showed that once appropriate polyubiquitin chains begin to accumulate on PML, they are very quickly recognised by p97, then turned over (Jaffray et al 2023 *J Cell Biol.*;222). Specifically, we found that to preserve ubiquitin conjugates associated with WT PML, cells need to be treated with a p97 inhibitor. Therefore, it is not surprising that ubiquitin shows only a modest increase with arsenic treatment when p97 is not inhibited. We have added text to the results section to clarify this point for readers.

4. Pearson Correlation Analysis for Figure 3B. Conducting a Pearson correlation coefficient analysis between SUMO1 and PML, and between SUMO2 and PML in the immunofluorescence experiments would enhance the clarity and quantitative analyses of the data.

We tried the suggested correlation analysis but this was unsuccessful owing to the non-uniform nature of PML bodies and the observation that while SUMO can be detected within PML bodies, it does not precisely colocalise with PML. Work from our own and other labs (Hattersley N, et al. 2011 *Mol Biol Cell*. 22:78-90; Lang M, et al 2010. *J Cell Sci*. 123:392-400) shows that PML forms an outer 'shell' with SUMO being localised to the core within. We therefore reverted to the manual colocalization analysis used throughout the manuscript. While this also has limitations, it provided more reliable data than the correlation analysis. We have included a summary of the data in new Fig 3. Overall, we think that the immunofluorescence along with the immunoblot and proteomic data make a very strong case for the differential recruitment of SUMO to PML bodies revealed by this work.

5. In Vitro Ubiquitination Assays (Line 524, Figure 7E). The in vitro ubiquitination assays indicate that 4xSUMO1 is preferentially ubiquitinated in the presence

of TOPORS, while both 4xSUMO1 and 4xSUMO2 are modified by RNF4. These observations, though intriguing, raise mechanistic questions. For example, what is the precise nature of the SUMO1 or mixed SUMO2/3-SUMO1 chains recognized by TOPORS? The use of 4xSUMO1 in vitro has not been confirmed on PML, leaving this point unresolved.

With respect to mixed SUMO chains, we refer to our response to reviewer 1 point 2. The work presented in the manuscript shows at least 13 different SUMO-SUMO linkages can be found associated with PML, and their length may vary from two to dozens of SUMO molecules. This degree of complexity means a thorough study of mixed SUMO chain binding by TOPORS is a significant undertaking. We think the present findings represent a significant advance on what was previously known, and while it does raise more questions, we prefer to reserve further characterisation to a future study.

6. TOPORS Colocalization Analysis for Figure 7. Including an immunofluorescence analysis of TOPORS colocalization across all three PML variants, similar to the RNF4 colocalization shown in Figure 3B, would be beneficial. Demonstrating TOPORS and PML colocalization in WT and SUMO1 ^{-/-} cells would further validate the selective recruitment of TOPORS.

We agree these would be excellent experiments but despite multiple attempts, we have been unable to execute them successfully. TOPORS is a very low copy number protein and the available antibodies for TOPORS do not work well in fluorescence experiments. Furthermore, our own attempts to generate good antibodies (in rabbits, sheep and chickens) have been unsuccessful. We have reservations over the use of TOPORS over-expression for these studies and would rather leave them for such times as the appropriate reagents are available. We think that the evidence in the manuscript strongly supports the idea TOPORS is a SUMO1-preferential E3 ligase involved in PML degradation, and hope the reviewer agrees that these specific experiments are not crucial to publication.

7. Clarifying TOPORS and RNF4 Cooperation. The study does not fully elucidate how TOPORS and RNF4 cooperate or compensate for each other in PML degradation. Additional experiments detailing their individual contributions and potential compensatory mechanisms would strengthen the section 'L217F PML fails to recruit TOPORS in response to arsenic'.

This work and others have shown RNF4 and TOPORS cooperate but are non-redundant. We agree it is intriguing mechanistically how multiple E3 ligases can be required for degradation of one protein, but it is not clear which simple experiments could be done to resolve this. In fact, this is the objective of my new CRUK programme grant, (which will keep 4 postdocs busy for 5 years). For this publication we'd prefer to keep the message simple, with the take-home message being that two ubiquitination pathways; TOPORS-SUMO1 and RNF4-SUMO2 converge on PML, and are required for its degradation

Minor points:

1. Line 242, These SUMO-ubiquitin species released by SENP1 were also detected using a ubiquitin antibody (Fig. S3B).

This has been corrected

2. Line 350 and Figure 4B, could the authors clarify whether the changes in relative intensity correspond to aggregate numbers for all ubiquitin or SUMO1/2/3 sites, considering that modifications can take place at multiple residues?

Fig 4B shows the relative intensity of the sum of all unmodified peptides and is therefore a proxy for protein abundance in the samples. We have clarified this in the figure legend.

3. Loading Controls: Several Western blots lack loading controls. Including these would improve the reliability of the data presented.

YFP-PML is the most appropriate control blot for YFP-PML purifications and is included in all figures as necessary. We have included control blots or highlighted non-specific bands for all other blots to evaluate loading.

February 17, 2025

RE: JCB Manuscript #202407133R

Ronald Hay
University of Dundee

Dear Prof. Hay,

Thank you for submitting your revised manuscript entitled "PML Mutants from Arsenic-Resistant Patients Reveal SUMO1-TOPORS and SUMO2-RNF4 Degradation Pathways." We would be happy to publish your paper in JCB pending final revisions necessary to meet our formatting guidelines (see details below).

A. MANUSCRIPT ORGANIZATION AND FORMATTING:

1) Text limits: Character count for Articles is < 40,000, not including spaces. Count includes title page, abstract, introduction, results, discussion, and acknowledgments. Count does not include materials and methods, figure legends, references, tables, or supplemental legends.

2) Figure formatting: Articles may have up to 10 main text figures. Scale bars must be present on all microscopy images, including inset magnifications. Molecular weight or nucleic acid size markers must be included on all gel electrophoresis. Please avoid pairing red and green for images and graphs to ensure legibility for color-blind readers. If red and green are paired for images, please ensure that the particular red and green hues used in micrographs are distinctive with any of the colorblind types. If not, please modify colors accordingly or provide separate images of the individual channels.

3) Statistical analysis: Error bars on graphic representations of numerical data must be clearly described in the figure legend. The number of independent data points (n) represented in a graph must be indicated in the legend. Please, indicate whether 'n' refers to technical or biological replicates (i.e. number of analyzed cells, samples or animals, number of independent experiments). If independent experiments with multiple biological replicates have been performed, we recommend using distribution-reproducibility SuperPlots (please see Lord et al., JCB 2020) to better display the distribution of the entire dataset, and report statistics (such as means, error bars, and P values) that address the reproducibility of the findings.

Statistical methods should be explained in full in the materials and methods. For figures presenting pooled data the statistical measure should be defined in the figure legends. Please also be sure to indicate the statistical tests used in each of your experiments (both in the figure legend itself and in a separate methods section) as well as the parameters of the test (for example, if you ran a t-test, please indicate if it was one- or two-sided, etc.). Also, if you used parametric tests, please indicate if the data distribution was tested for normality (and if so, how). If not, you must state something to the effect that "Data distribution was assumed to be normal but this was not formally tested."

4) Materials and methods: Should be comprehensive and not simply reference a previous publication for details on how an experiment was performed. Please provide full descriptions (at least in brief) in the text for readers who may not have access to referenced manuscripts. The text should not refer to methods "...as previously described." Please also indicate the acquisition and quantification methods for immunoblotting/western blots.

5) For all cell lines, vectors, constructs/cDNAs, etc. - all genetic material: please include database / vendor ID (e.g. Addgene, ATCC, etc.) or if unavailable, please briefly describe their basic genetic features, even if described in other published work or gifted to you by other investigators (and provide references where appropriate). Please be sure to provide the sequences for all of your oligos: primers, si/shRNA, RNAi, gRNAs, etc. in the materials and methods. You must also indicate in the methods the source, species, and catalog numbers/vendor identifiers (where appropriate) for all of your antibodies, including secondary. If antibodies are not commercial, please add a reference citation if possible.

6) Microscope image acquisition: The following information must be provided about the acquisition and processing of images:

- Make and model of microscope
- Type, magnification, and numerical aperture of the objective lenses
- Temperature
- Imaging medium

- e. Fluorochromes
- f. Camera make and model
- g. Acquisition software
- h. Any software used for image processing subsequent to data acquisition. Please include details and types of operations involved (e.g., type of deconvolution, 3D reconstitutions, surface or volume rendering, gamma adjustments, etc.).

7) References: There is no limit to the number of references cited in a manuscript. References should be cited parenthetically in the text by author and year of publication. Abbreviate the names of journals according to PubMed.

8) Supplemental materials: Articles generally may have up to 5 supplemental figures and 10 videos. You currently exceed this limit but, in this case, we will be able to give you the extra space. The text in Figure S2E/F is somewhat small and may not be legible in the final figure, please try to enlarge this. Tables, like figures, should be provided as individual, editable files. A summary of all supplemental material should appear at the end of the Materials and methods section. Please include one brief sentence per item.

9) Video legends: Should describe what is being shown, the cell type or tissue being viewed (including relevant cell treatments, concentration and duration, or transfection), the imaging method (e.g., time-lapse epifluorescence microscopy), what each color represents, how often frames were collected, the frames/second display rate, and the number of any figure that has related video stills or images.

10) eTOC summary: A ~40-50 word summary that describes the context and significance of the findings for a general readership should be included on the title page. The statement should be written in the present tense and refer to the work in the third person. It should begin with "First author name(s) et al..." to match our preferred style.

11) Conflict of interest statement: JCB requires inclusion of a statement in the acknowledgements regarding competing financial interests. If no competing financial interests exist, please include the following statement: "The authors declare no competing financial interests." If competing interests are declared, please follow your statement of these competing interests with the following statement: "The authors declare no further competing financial interests."

12) A separate author contribution section is required following the Acknowledgments in all research manuscripts. All authors should be mentioned and designated by their first and middle initials and full surnames. We encourage use of the CRediT nomenclature (<https://casrai.org/credit/>).

13) ORCID IDs: ORCID IDs are unique identifiers allowing researchers to create a record of their various scholarly contributions in a single place. Please note that ORCID IDs are required for all authors. At resubmission of your final files, please be sure to provide your ORCID ID and those of all co-authors.

14) JCB requires authors to submit Source Data used to generate figures containing gels and Western blots with all revised manuscripts. This Source Data consists of fully uncropped and unprocessed images for each gel/blot displayed in the main and supplemental figures. For assays performed using capillary electrophoresis and/or immunoassay-based detection, authors should instead provide the electropherogram graph(s) for each experiment, plotting fluorescence/chemiluminescence intensity vs. molecular weight/size. Since your paper includes cropped gel and/or blot images, please be sure to provide one Source Data file for each figure gels, blots, and/or capillary electrophoresis assays along with your revised manuscript files. File names for Source Data figures should be alphanumeric without any spaces or special characters (i.e., SourceDataF#, where F# refers to the associated main figure number or SourceDataFS# for those associated with Supplementary figures). For traditional gels and blots, the lanes of the gels/blots should be labeled as they are in the associated figure, the place where cropping was applied should be marked (with a box), and molecular weight/size standards should be labeled wherever possible. For capillary electrophoresis assays, each trace in the graph should be color-coded and labeled to indicate which protein, gene, or sample is being measured (please try to avoid red/green combinations to accommodate our color-blind readers).

Source Data files will be directly linked to specific figures in the published article. Source Data Figures should be provided as individual PDF files (one file per figure). Authors should endeavor to retain a minimum resolution of 300 dpi or pixels per inch. Please review our instructions for export from Photoshop, Illustrator, and PowerPoint here: <https://rupress.org/jcb/pages/submission-guidelines#revised>

15) Journal of Cell Biology now requires a data availability statement for all research article submissions. These statements will be published in the article directly above the Acknowledgments. The statement should address all data underlying the research presented in the manuscript. Please visit the JCB instructions for authors for guidelines and examples of statements at (<https://rupress.org/jcb/pages/editorial-policies#data-availability-statement>).

B. FINAL FILES:

Thank you for your attention to these final processing requirements. Please revise and format the manuscript and upload materials within 7 days. If you need an extension for whatever reason, please let us know and we can work with you to determine a suitable revision period.

Thank you for this interesting contribution, we look forward to publishing your paper in Journal of Cell Biology.

Sincerely,

Ira Mellman, PhD
Monitoring Editor
Journal of Cell Biology

Dan Simon, PhD
Scientific Editor
Journal of Cell Biology

Reviewer #1 (Comments to the Authors (Required)):

The authors have thoroughly revised their manuscript and my main concerns have been properly addressed.

Reviewer #2 (Comments to the Authors (Required)):

The authors have sufficiently addressed most reviewer concerns, either through additional data, clarification, or well-reasoned justifications for why certain experiments were not feasible. Partial resolution was presented for points 5 and 7. For point 5, the authors acknowledge the complexity of mixed SUMO chains and justify deferring further characterization to future studies. This leaves an open mechanistic question, but their justification aligns with the study's focus. For point 7, the authors clarify that their current study provides an initial framework and acknowledges the need for further mechanistic studies, which will be explored in future research. Obviously, this does not fully resolve the question, but it is a reasonable limitation given the study's scope.